# Provable Unlearning in Topic Modeling and Downstream Tasks

**Stanley Wei**\*, **Sadhika Malladi**\*, **Sanjeev Arora**
Princeton University
{stanley.wei, smalladi, arora}@princeton.edu

**Amartya Sanyal**
University of Copenhagen
amsa@di.ku.dk

## Abstract

Machine unlearning algorithms are increasingly important as legal concerns arise around the provenance of training data, but verifying the success of unlearning is often difficult. Provable guarantees for unlearning are often limited to supervised learning settings. In this paper, we provide the first theoretical guarantees for unlearning in the pre-training and fine-tuning paradigm by studying topic models, simple bag-of-words language models that can be adapted to solve downstream tasks like retrieval and classification. First, we design a provably effective unlearning algorithm for topic models that incurs a computational overhead independent of the size of the original dataset. Our analysis additionally quantifies the deletion capacity of the model – *i.e.,* the number of examples that can be unlearned without incurring a significant cost in model performance. Finally, we formally extend our analyses to account for adaptation to a given downstream task. In particular, we design an efficient algorithm to perform unlearning after fine-tuning the topic model via a linear head. Notably, we show that it is easier to unlearn pre-training data from models that have been fine-tuned to a particular task, and one can unlearn this data without modifying the base model.

## 1 Introduction

Modern-day machine learning has shifted from single-stage supervised learning on manually constructed datasets to a paradigm in which models are pre-trained and subsequently fine-tuned (Bommasani et al., 2022). In this setting, a model initially learns a good representation of the data using a self-supervised objective on a large unstructured corpus. The resulting pre-trained model is later adapted to solve specific tasks for which it is difficult or costly to curate a large dataset. This blueprint has yielded strong performance in text (*e.g.,* Devlin et al., 2019; Brown et al., 2020), vision (*e.g.,* Oquab et al., 2024; He et al., 2022), and multimodal (*e.g.,* Radford et al., 2021; Zhai et al., 2023) settings. It is well-known that the scale of the pre-training data is strongly correlated with the final performance of the model (Hoffmann et al., 2022), leading to the construction of larger datasets via broad internet scrapes (Gao et al., 2020; Schuhmann et al., 2022; Soldaini et al., 2024; Penedo et al., 2023). Such datasets have been found to often inadvertently include private, sensitive, and unsafe data (Birhane et al., 2021; Longpre et al., 2024; He et al., 2024).

Unsafe data can generally degrade model performance and introduce biases, making the model less useful for various applications (McKenna et al., 2023; Birhane & Prabhu, 2021; Choenni et al., 2021; Naous et al., 2024). Using private and sensitive data, even unknowingly, poses legal risks (Bommasani et al., 2022; Henderson et al., 2023). In particular, recent works have shown that models can memorize and thus permit the extraction of training data (Somepalli et al., 2023; Carlini et al., 2021; 2023). Moreover, one may be requested to remove data in accordance with GDPR's *right to be forgotten* (European Parliament & Council of the European Union), or as part of a copyright-related lawsuit (*Tremblay v. OpenAI, Inc.,*, 2023; *DOE 1 v. GitHub, Inc.*, N.D. Cal. 2022).

Therefore, there is great empirical interest in developing machine unlearning algorithms that can surgically remove portions of the training data from an already learned model without harming performance. The gold standard for machine unlearning is for the model to behave as though it had

---
\*Equal contribution.

never been trained on that datapoint (Cao & Yang, 2015). As it is often undesirable to completely retrain models, especially as they grow larger, many works have proposed computationally cheaper heuristics for solving this problem (*e.g.*, Jang et al., 2023; Foster et al., 2024; Kurmanji et al., 2023; Zhang et al., 2024b; Eldan & Russinovich, 2023; Gandikota et al., 2023). In the absence of theoretical guarantees, it is common to use empirics to measure the success of these algorithms. However, recent works have shown that such evaluations often overestimate the success of these unlearning methods (Hayes et al., 2024; Shi et al., 2024; Maini et al., 2024) and thus it has proven difficult to confidently ascertain whether the proposed methods meet the necessary compliance standards. In this context, it is highly desirable to design efficient unlearning algorithms with well-motivated guarantees that are salient to the pre-training and finetuning paradigm (Thudi et al., 2022; Lee et al., 2024).

While there are some instances of such algorithms for linear models (Guo et al., 2020; Izzo et al., 2021; Mahadevan & Mathioudakis, 2023), general convex models (Ullah et al., 2021; Sekhari et al., 2021; Neel et al., 2021), Bayesian models (Nguyen et al., 2020), and GANs (Liu et al., 2024), there are no works on the paradigm of pre-training and fine-tuning algorithms. One of the most classical such algorithms is topic modeling (Hofmann et al., 1999; Blei et al., 2003; Blei & Lafferty, 2006; Li & McCallum, 2006), which can also be thought of as the simplest language model. *In this paper, we present the first provably effective and efficient unlearning algorithms for topic models.*

Topic models are generally pre-trained to extract latent structure (i.e., a small set of underlying topics) from a large corpus of documents. This feature extractor is then used for a variety of downstream applications, including retrieval, classification, and recommendation (Boyd-Graber et al., 2017). Despite their simplicity, topic models can be used to effectively solve many real-world natural language problems — see a survey in Churchill & Singh (2022).

## 1.1 OVERVIEW OF RESULTS

We focus on the setting in Arora et al. (2012b), because it admits an efficient learning algorithm with provable guarantees (Arora et al., 2012a). The corpus is assumed to contain $r$ underlying topics, where each topic defines a distribution over words. Let $\mathcal{D}$ be a distribution over topic distributions. Then, each document $d$ is generated by sampling a topic distribution $W_d \sim \mathcal{D}$ over topics, and then sampling words according to $W_d$.

The dataset of $m$ documents is a matrix $\boldsymbol{M} \in \mathbb{R}^{n \times m}$, where $\boldsymbol{M}$ permits a non-negative matrix factorization $\boldsymbol{M} = \boldsymbol{A}^* \boldsymbol{X}$. Here, $\boldsymbol{A}^* \in \mathbb{R}^{n \times r}$ is the distribution of words in each of the $r$ unknown underlying topics, and $\boldsymbol{X} \in \mathbb{R}^{r \times m}$ is the sampled distribution of topics in each document. In particular, $\boldsymbol{A}^\star, \boldsymbol{X}$ have columns on the probability simplex. We seek to learn the embedding function $\boldsymbol{A}^*$ and the topic-topic covariance $\boldsymbol{R}^\star = \mathbb{E}_{\mathcal{D}}[\boldsymbol{X} \boldsymbol{X}^\top]$.

To derive provable guarantees on the success of unlearning, we adapt the notion of $(\epsilon, \delta)$-unlearning introduced in Sekhari et al. (2021) to the topic modeling setting. The unlearned model is required to behave indistinguishably from a model that was retrained on the modified dataset. We define a notion of *utility-preserving unlearning* that combines this condition with an analysis on the *deletion capacity* – i.e., the number of datapoints that can be unlearned without performance degradation (Definition 4). We now state our main result on utility-preserving unlearning in topic models.

**Main Result 1** (Informal version of Theorem 2). Suppose we trained a topic model $\boldsymbol{A}^S, \boldsymbol{X}^S$ on a training set $S$ containing $m$ documents. Algorithm 1 can perform utility-preserving unlearning of

$$m_U = \tilde{\mathcal{O}}\left(\frac{m}{r^2 \sqrt{nr}}\right)$$

documents from the pre-trained topic model, where $\tilde{\mathcal{O}}(\cdot)$ hides constants depending on the learning and unlearning algorithm.

To adapt a topic model to a downstream topic classification task, we learn a head $\boldsymbol{w} \in \mathbb{R}^r$ on top of $\boldsymbol{A}$ to minimize a strongly convex loss function (Definition 2). When $\boldsymbol{A}$ and $\boldsymbol{w}$ are both released, one would necessarily have to first unlearn from $\boldsymbol{A}$, which makes unlearning just as hard as it was in pre-training (Theorem 3). This setting is rather unrealistic, because there is no obvious case in which one would want to use $\boldsymbol{w}$ without $\boldsymbol{A}$ or vice versa. We thus advocate for viewing fine-tuned model $\boldsymbol{B} = \boldsymbol{A}\boldsymbol{w}$ as a whole i.e. it is not allowed to access outputs of $\boldsymbol{A}$ solely, and we show that it is easier to perform utility-preserving unlearning of pre-training data in this case.

**Main Result 2** (Informal version of Theorem 4)**.** After adapting the model to a downstream task (Definitions 1 and 2), Algorithm 2 can perform utility-preserving unlearning of $\tilde{\Omega}\left(\frac{mq}{r\sqrt{nr}}\right)$ documents, where $q \in [1/r, 1]$ is a task-dependent quantity, without modifying the base model $\boldsymbol{A}$. Simpler downstream tasks have a larger $q$, increasing the separation from the pre-training result.

We demonstrate that our unlearning algorithms run substantially faster than retraining the model (Table 1). Overall, our results imply the following takeaways in the context of topic models — (1) It is possible to effectively and efficiently unlearn datapoints from a pre-trained model without retraining it (Algorithm 1 and Theorem 2), (2) One can effectively unlearn more pre-training data from a model that has been adapted to a downstream task without harming the utility of the base and fine-tuned models (Theorem 4), and (3) One can unlearn pre-training data from a fine-tuned model without modifying the base model (Algorithm 2 and Theorem 4).

## 2 TOPIC MODELS

As we previously discussed, topic models can be considered as one of the simplest language models that one can pre-train in a self-supervised fashion and later fine-tune for other language-related tasks. This pipeline mirrors the modern-day paradigm of pre-training large language models to build a general understanding of natural language and later fine-tuning them to solve a variety of tasks ranging from classification to code generation.

### 2.1 PROBLEM DESCRIPTION

Topic modeling is a classical, bag-of-words method to discover structure in a corpus of documents (Hofmann et al., 1999). One assumes that each document contains a convex combination of topics, each of which can be described in terms of a distribution over the vocabulary. Different assumptions on the structure of this distribution and the topics have yielded a variety of topic modeling methodologies (Blei & Lafferty, 2006; Li & McCallum, 2006) – perhaps most famous among these is the latent Dirichlet allocation (LDA, Blei et al. (2003)). Many early works established the statistical learnability of topic models under such assumptions, but the learning algorithms generally were not efficient in real-world settings (Arora et al., 2012b; Recht et al., 2012).

Our paper focuses on the setting in Arora et al. (2012b), for which Arora et al. (2012a) provided an empirically efficient learning algorithm. The dataset consists of a set of $m$ documents $d_1, ..., d_m$, where each document contains $L$ words from a vocabulary $\mathcal{V}$ with $|\mathcal{V}| = n$.[1] The corpus contains $r$ different underlying topics, each of which defines a distribution over words. Each word in document $d$ is generated by: (1) sampling a distribution over topics $W_d \sim \mathcal{D}$, and then (2) sampling $L$ words independently according to $W_d$.

We represent the corpus as a matrix $\boldsymbol{M} \in \mathbb{R}^{n \times m}$, where $\boldsymbol{M}$ permits a non-negative matrix factorization $\boldsymbol{M} = \boldsymbol{A}^\star \boldsymbol{X}$. Here, $\boldsymbol{A}^\star \in \mathbb{R}^{n \times r}$ is the distribution of words in each of the $r$ topics, $\boldsymbol{X} \in \mathbb{R}^{r \times m}$ is the distribution of topics in each document, and hence $\boldsymbol{M}$ is the distribution of words in each document. While there are several algorithms for learning the feature extractor $\boldsymbol{A}^\star$, it is well-known that it is hard to recover $\boldsymbol{X}$ exactly (Arora et al., 2012b). Instead, it is desirable to learn how the topics co-occur together, denoted as $\boldsymbol{R}^\star = \mathbb{E}_{\mathcal{D}}[\boldsymbol{X}\boldsymbol{X}^\top]$. This quantity is termed the *topic-topic covariance*. Further discussion of this has been included in Appendix A.

The topic modeling setting generally determines $\mathcal{D}$ (e.g., in LDA, $\mathcal{D}$ is a Dirichlet distribution). In order to recover $\boldsymbol{A}^*$ and $\boldsymbol{R}^*$ efficiently and accurately from an observed corpus $\boldsymbol{M} \sim \mathcal{D}$, we need to make the following assumption on the underlying data distribution.

**Assumption 1** ($p$-separability, Arora et al. (2012b))**.** *The topic matrix $\boldsymbol{A}^\star$ is $p$-separable for $p > 0$ if for every topic $k \in [r]$, there exists a word $i \in [n]$ such that $\boldsymbol{A}_{i,k}^* \geq p$ and $\boldsymbol{A}_{i,k'}^* = 0$ for all $k' \neq k$. Such words are called* anchor words.

Without this separability assumption, maximum likelihood estimation of a topic model is NP-hard (Arora et al., 2012b). Assumption 1 requires that $\boldsymbol{A}^\star$ contains a diagonal matrix, up to row

---

[1]Without loss of generality, we assume $L = 2$. The same asymptotic results hold for constant sized $L$, and it is straightforward to modify our analysis to account for this.

permutations; intuitively, the appearance of an anchor word in a document perfectly indicates the document has nonzero probability of the corresponding topic. As we will detail in Section 4, this observation inspires a two-phase learning algorithm, whereby one first approximates the anchor words for each topic and then leverages them to identify patterns among the topics.

## 2.2 DOWNSTREAM ADAPTATION

Topic models are frequently trained on a general corpus, and the embeddings can be later used to classify documents. The classification problem usually involves only a subset of topics. For example, after training a topic model on a large corpus of news articles with diverse topics (e.g., sports, politics, technology, finance, etc.), one relevant downstream task is to classify the subject of a given news article as sports or politics. We formalize the topic classification task below.

**Definition 1** (Topic Classification Task). A topic classification task $\mathcal{T} = (\mathbb{T}_{\text{clf}}, \boldsymbol{w}^{\star})$ is defined by a subset of topics $\mathbb{T}_{\text{clf}} \subset [r]$ on which the task is defined and a ground-truth labelling vector $\boldsymbol{w}^{\star} \in \mathbb{R}^r$ with bounded norm. Importantly, $\boldsymbol{w}^{\star}$ only has non-zero coordinates in the positions corresponding to $\mathbb{T}_{\text{clf}}$.

The classification task is defined on the latent features of a given document, so it is necessary to first identify the salient topics as they occur in the text. Fitting a topic model to the corpus yields such a feature extractor $\boldsymbol{A}$ that embeds a document into the $r$-dimensional topic space. In order to adapt a topic model to a particular classification task, we perform head tuning on the feature extractor $\boldsymbol{A}$.

**Definition 2** ($\tau$-Head Tuning). For a given labelled document classification dataset $\mathbb{D}_{\text{clf}} = \{(d_i, y_i)\}$ representing a topic classification task $\mathcal{T}$, embed each document $d_i$ as a vector $\boldsymbol{x}_i \in \mathbb{R}^n$ containing the word counts in the document. To perform head tuning on a pre-trained topic model $\boldsymbol{A}$, we learn $\boldsymbol{w}_\tau \in \mathbb{R}^r$ that is a $\tau$-optimal point of

$$\ell_{\mathcal{T}}(\boldsymbol{w}; \boldsymbol{A}) = \frac{1}{|\mathbb{D}_{\text{clf}}|} \sum_{(\boldsymbol{x},y) \in \mathbb{D}_{\text{clf}}} f(\boldsymbol{x}^\top \boldsymbol{A} \boldsymbol{w}, y)$$

where $\ell_{\mathcal{T}}$ is strongly convex in $\boldsymbol{w}$. In other words, $\boldsymbol{w}_\tau$ satisfies $\ell_{\mathcal{T}}(\boldsymbol{w}_\tau; \boldsymbol{A}) - \min_{\boldsymbol{w}} \ell_{\mathcal{T}}(\boldsymbol{w}; \boldsymbol{A}) \leq \tau$.

One example of $f$ is the logistic loss with $\ell_2$ regularization. For ease of exposition, we primarily consider binary classification tasks, but we point out that the definition can extend to multi-class tasks solved via the one-vs-all scheme (Rifkin & Klautau, 2004).

We note that head tuning, also referred to as linear probing, is a simpler adaptation technique than fine-tuning $\boldsymbol{A}$ alongside $\boldsymbol{w}$. Nonetheless, recent works on popular language models have demonstrated that head tuning can substantially improve the ability of general pre-trained language models to solve complex classification tasks (Malladi et al., 2023a;b). Head tuning thus serves as a convenient yet effective adaptation method that avoids updating the pre-trained model, which is often desirable. For example, if a single pre-trained model needs to be separately adapted to solve many different tasks, then it is desirable to minimize the number of parameters that are fine-tuned to minimize the memory needed to store all of the adapted models.[2]

## 3 UNLEARNING

As mentioned previously, there is increased interest in machine unlearning due to the growing scale of modern datasets and the difficulty of manually inspecting each datapoint. Theoretically, the gold standard for unlearning is that the model should behave identically to one that was trained without the datapoint in its corpus (Cao & Yang, 2015). We first define what it means for two models $\theta_1, \theta_2 \in \Theta$ to behave *almost* identically, where $\Theta$ denotes the parameter space of a hypothesis class. Due to randomness in learning, $\theta_1$ and $\theta_2$ are random variables.

**Definition 3** (($\epsilon, \delta$)-indistinguishable models, Dwork et al. (2014)). Two models denoted by random variables $\theta_1, \theta_2 \in \Theta$ are ($\epsilon, \delta$)-indistinguishable if for all possible subsets of models $T \subseteq \Theta$,

$$\Pr(\theta_1 \in T) \leq e^\epsilon \Pr(\theta_2 \in T) + \delta \quad \text{and} \quad \Pr(\theta_2 \in T) \leq e^\epsilon \Pr(\theta_1 \in T) + \delta$$

---

[2] This motivation has driven widespread development and adoption of parameter-efficient fine-tuning methods for large language models. Liu et al. (2021) contains a survey of such techniques.

We adapt the definitions from Sekhari et al. (2021) to the topic modeling setting. A learning algorithm $\mathcal{A}$ takes in a set of $m$ documents, denoted as $S$, and returns a topic model $\theta = (\boldsymbol{A}, \boldsymbol{R})$. Analogously, an unlearning algorithm $\mathcal{U}$ takes in the learned topic model $\theta$, a set of documents to unlearn $S_f \subseteq S$, some statistics on the training set $T(S)$, and outputs a model. The set of datapoints to unlearn $S_f$ is often referred to as the *forget set*. With this in mind, we now define a notion of utility-preserving unlearning, whereby the unlearning algorithm needs to not only effectively simulate retraining the model from scratch but also maintain the model's performance.

**Definition 4** (Utility-preserving $(\epsilon, \delta)$-Unlearning with Deletion Capacity). Let $m \in \mathbb{N}$ be a constant that depends on the topic modeling distribution $\mathcal{D}$ satisfying Assumption 1. For any training dataset $S \stackrel{\text{i.i.d.}}{\sim} \mathcal{D}$ of size at least $m$, and $\epsilon, \delta > 0$, we say that a pair of learning and unlearning algorithms $(\mathcal{A}, \mathcal{U})$ performs *utility-preserving unlearning with deletion capacity* $T_{\epsilon,\delta}^{\mathcal{A},\mathcal{U}}(m)$ if

1. With probability at least 0.9 over draws from $\mathcal{D}$, for any forget set $S_f \subseteq S$ of size at most $T_{\epsilon,\delta}^{\mathcal{A},\mathcal{U}}(m)$, model trained on $S \setminus S_f$ is indistinguishable from that resulting from unlearning with $\mathcal{U}$. Here, $\stackrel{\epsilon,\delta}{\approx}$ denotes $(\epsilon, \delta)$-indistinguishability.

$$\mathcal{U}(S_f, \mathcal{A}(S), T(S)) \stackrel{\epsilon,\delta}{\approx} \mathcal{U}(\emptyset, \mathcal{A}(S \setminus S_f), T(S \setminus S_f))$$

2. Even for an adversarially chosen $S_f$, the unlearned model does not suffer a large performance degradation. Formally,

$$\mathbb{E}_{\mathcal{A},\mathcal{U}} \left[ \max_{|S_f| \leq T_{\epsilon,\delta}^{\mathcal{A},\mathcal{U}}(m)} h(\mathcal{U}(S_f, \mathcal{A}(S), T(S))) - h^\star \right] \leq 0.01$$

where $h : \Theta \to \mathbb{R}$ is the loss of the topic model, and $h^\star = \min_{w \in \mathcal{W}} h(w)$ is the irreducible loss.

The above definition can be applied to both the pre-training and the downstream adaptation stages of training a topic model. Of particular note is that (1) does not guarantee (2), since the former only concerns indistinguishability between the unlearned and retrained models, while the latter is a statement about utility preservation. Moreover, unless $T(S)$ contains the entire dataset, we note that the unlearning algorithm $\mathcal{U}$ cannot be as simple as retraining the model. In this paper, we will design an unlearning algorithm for topic models that satisfies this definition of provable unlearning, and the number of statistics $T(S)$ will not depend on the initial dataset size $m$.

To show $(\epsilon, \delta)$-indistinguishability, we utilize the Gaussian mechanism, a classic tool from differential privacy. Given a particular function, the Gaussian mechanism essentially prescribes how much noise one must add to the output in order for the input to be indistinguishable from a similar one. The guarantee of the Gaussian mechanism is described in the following lemma.

**Lemma 1** (Gaussian Mechanism, Dwork et al. (2014)). *Let $f$ be an arbitrary $d$-dimensional function, and define its $\ell_2$-sensitivity to be $\Delta_2 f := \max\limits_{adjacent\ x,y} \|f(x) - f(y)\|_2$. Then, for $c^2 > 2\log\frac{1.25}{\delta}$, the Gaussian mechanism with parameter $\sigma \geq c\Delta_2 f / \epsilon$ is $(\epsilon, \delta)$-differentially private.*

In our case, we define adjacent inputs (i.e., training datasets) as the case where $y$ is a superset of $x$.

## 4 LEARNING AND UNLEARNING TOPIC MODELS

In this section, we present the learning and unlearning algorithms and guarantees for topic models.

**Notation.** We use $\boldsymbol{A}^\star$ to refer to the ground-truth topic model, $\boldsymbol{A}^S$ to refer to a topic model trained on $S$, and $\boldsymbol{A}^F$ to denote a topic model retrained with the forget set removed $S \setminus S_f$. We also use $\bar{\boldsymbol{A}}$ to denote the unlearned topic model before applying the Gaussian mechanism and $\tilde{\boldsymbol{A}}$ to denote the model after the mechanism is applied. Analogous notations are used for $\boldsymbol{R}$.

### 4.1 LEARNING ALGORITHM AND GUARANTEES

Per Arora et al. (2012a), the learning algorithm $\mathcal{A}_{\text{base}}$ takes in a corpus of documents $S = \{d_1, ..., d_m\}$ and consists of the following three phases to learn a topic model $\theta = (\boldsymbol{A}^S, \boldsymbol{R}^S)$.

1. **Measure the word co-occurrences.** Compute the word co-occurrence matrix $\boldsymbol{Q} \in \mathbb{R}^{n \times n}$, where $Q_{ij}$ is the number of times word $i$ appears in the same document as word $j$. We also compute $\bar{\boldsymbol{Q}}$, which normalizes the rows of $\boldsymbol{Q}$ to sum to 1. A detailed discussion of the construction of $\boldsymbol{Q}$ and its relationship to the factorization $\boldsymbol{M} = \boldsymbol{A}^\star \boldsymbol{X}$ is included in Appendix A.

2. **Identify the anchor words** $P$. Recall that in order to be able to learn topic models efficiently, there must exist a set of anchor words $P$ with $|P| = r$, and each anchor word must appear exclusively in a single topic (Assumption 1). This subroutine uses $\bar{\boldsymbol{Q}}$ to approximately identify the $r$ anchor words $P$.

3. **Learn the feature extractor $\boldsymbol{A}^S$ and the topic-topic covariance $\boldsymbol{R}^S$.** The algorithm uses the anchor words $P$ and the word co-occurrences $\bar{\boldsymbol{Q}}$ to learn $\boldsymbol{A}^S$ and $\boldsymbol{R}^S$. Each word is expressed as a convex combination of anchor words, and thus, topics. With appropriate normalization and by cross-referencing information with the co-occurrence matrix, one can recover $\boldsymbol{A}^\star, \boldsymbol{R}^\star$ in the infinite data limit.

We sketch how this algorithm recovers the ground truth $\boldsymbol{A}^\star, \boldsymbol{R}^\star$ when one has infinitely many documents in Appendix A. Arora et al. (2012a) gives the following finite-document guarantee.

**Theorem 1** (Learning Guarantee). *Running $\mathcal{A}_{base}$ on a dataset $S$ of size $m$, where $m$ is at least*

$$\max\left\{ \mathcal{O}\left( \frac{ar^3 \log n}{L(\gamma p)^6 \epsilon_0} \right), \mathcal{O}\left( \frac{a^3 r^3 \log n}{L\epsilon_0^3 (\gamma p)^4} \right), \mathcal{O}\left( \frac{r^2 \log r}{L\epsilon_0^2} \right) \right\}$$

*recovers $\boldsymbol{A}^S$ and $\boldsymbol{R}^S$ with entrywise additive error up to $\epsilon_0$ from the ground truth $\boldsymbol{A}^\star, \boldsymbol{R}^\star$, respectively. Here, $a$ is the topic imbalance parameter, and $\gamma$ is the condition number of the ground truth $\boldsymbol{R}^\star$. Formally, we have $a = \max_{i,j \in [r]} \Pr_{\mathcal{D}}[z = i] / \Pr_{\mathcal{D}}[z = j]$.*

**Approximating the anchor words.** We defer a precise description of the anchor word identification algorithm to Appendix A and instead focus here on the intuitions driving its design and the guarantees we will use throughout the paper. First, we note the relationship between $\bar{\boldsymbol{Q}}$ and the set of anchor words. If we had infinitely many documents, then the convex hull of the rows in $\bar{\boldsymbol{Q}}$ will be a simplex with vertices corresponding to the anchor words, because each anchor word corresponds to a topic, and each topic prescribes a distribution over words. However, in the finite document setting, each row of $\bar{\boldsymbol{Q}}$ only approximates their expected value, and so one must approximate the vertices of a convex hull when given access to a perturbation of the points that define it.

We start by requiring that each topic is distinctly different from any mixture on the other topics. Formally, this requires that the simplex is robust, in that each vertex (i.e., anchor word) is sufficiently far from any combination of the other topics. Most topic modeling settings define lower bounds on the robustness of the simplex. By a result in Arora et al. (2012b), the simplex defined by the $r$ anchor word rows of the population $\bar{\boldsymbol{Q}}$ is $\gamma p$-robust. We can now define exactly the sense in which a $\bar{\boldsymbol{Q}}$ computed on a finite dataset approximates the population co-occurrence matrix.

**Definition 5.** Let $\{a_i\}_{i=1}^n$ be a set of points whose convex hull $P$ is a simplex with vertices $\{v_i\}_{i=1}^r$. We say a set of $r$ points is $\epsilon$-close the vertex set $\{v_i\}_{i=1}^r$ if each of the $r$ points is $\epsilon$-close in $\ell_2$ distance to a different vertex in $P$. Moreover, we say that a simplex $P$ is $\beta$-robust if for every vertex $v$ of $P$, the $\ell_2$ distance between $v$ and the convex hull of the rest of the vertices as at least $\beta$.

In the context of this definition, $P$ corresponds to the ground truth convex hull, and the finite sample $\bar{\boldsymbol{Q}}$ can be seen as a perturbation to it. In particular, Arora et al. (2012a) used this to established a guarantee on the accuracy of anchor word recovery.

**Lemma 2** (Approximation Guarantee on Anchor Words). *Suppose each row of $\bar{\boldsymbol{Q}}$ is at most $\delta$ distance away from the ground truth $\gamma p$-robust simplex $\bar{\boldsymbol{Q}}^\star$ in $\ell_2$ norm. If $20r\delta/(\gamma p)^2 < \gamma p$, then the set of anchor words found by the algorithm is $O(\delta/\gamma p)$-close to the ground truth anchor words.*

We now describe how to use the recovered approximate anchor words to learn the topic model.

**Learning the topic model from anchor words.** We are given the set of anchor words $P$, the word co-occurrence matrix $\boldsymbol{Q} \in \mathbb{R}^{n \times n}$, and the normalized co-occurrence matrix $\bar{\boldsymbol{Q}}$. Our goal is to use these quantities to learn $\boldsymbol{A} \in \mathbb{R}^{n \times r}$ and $\boldsymbol{R} \in \mathbb{R}^{r \times r}$. We will do so by first expressing each word $i \in [n]$ as a convex combination of the anchor words (and thus, the topics). In particular, for each word $i$, we learn the coefficients $\boldsymbol{C}_i \in \Delta_r$ as

$$\boldsymbol{C}_i = \underset{v \in \Delta_r}{\arg\min} \|\bar{\boldsymbol{Q}}_i - \boldsymbol{v}^\top \bar{\boldsymbol{Q}}_P\|^2$$

---

**Algorithm 1** Unlearning algorithm ($\mathcal{U}_{\text{base}}$)

---

**Input:** Forget set $S_f \subseteq S$, statistics $T(S)$ which include $\{C_i^S\}_{i=1}^n$, $Q^S$, $P$, normalization constants $p^S$

**Output:** Unlearned model $\tilde{A}, \tilde{R}$

Compute the updated co-occurrence matrix $Q^F$ by subtracting documents in $S_f$

Store the updated normalization constants $p^F = Q^F \mathbf{1}$

**for** $i$ in $1, \ldots, n$ **do**

    Newton step update on $C_i$'s:

$$\bar{C}_i^F \leftarrow C_i^S - H_{C_i^S}^{-1} \nabla \mathcal{L}(C_i^S, S \setminus S_f)$$

$$\bar{C}_i^F \leftarrow \text{proj}_{\Delta_r}(\bar{C}_i^F)$$

    where $\mathcal{L}(v, S \setminus S_f) := \|\bar{Q}_i^F - v^\top \bar{Q}_P^F\|^2$ and $H_{C_i^S} = \nabla^2 \mathcal{L}(C_i^S, S \setminus S_f)$

**end for**

$\bar{A}' = \text{diag}(p^F)\bar{C}$

$\bar{A} =$ column normalized $\bar{A}'$

$\bar{R} = \bar{A}^\dagger Q^F \bar{A}^{\dagger\top}$ where $\bar{A}^\dagger$ is the pseudoinverse of $\bar{A}$

Sample $\nu_A, \nu_R$ from normal distribution defined by Gaussian mechanism guarantee

$\tilde{A} =$ Project each column of $\bar{A} + \nu_A$ to $\Delta_n$.

$\tilde{R} =$ Project $\bar{R} + \nu_R$ onto the set of PSD matrices.

**return** The unlearned topic model $\tilde{A}, \tilde{R}$

---

where $\bar{Q}_P$ is the $P$ rows of $\bar{Q}$ corresponding to the anchor words. Arora et al. (2012a) showed the following approximation guarantee for $C_i$ compared to the ground-truth coefficients.

**Lemma 3.** *When $20r\delta/(\gamma p)^2 < \gamma p$, for every word $i$, $C_i$ has entrywise error $O(\delta/(\gamma p)^2)$ from $C_i^\star$.*

We then normalize this $C_i$ by the total number of co-occurrences that word $i$ is involved in. Note that the $C_i$ can be assembled into a matrix $C \in \mathbb{R}^{n \times r}$. We set $A$ to be $C$ after normalizing the columns sum to 1, since the columns represent the topic-conditioned distribution over the vocabulary. We finally compute $R = A^\dagger Q {A^\dagger}^\top$, where $A^\dagger$ denotes the pseudoinverse of $A$.

## 4.2 Unlearning Algorithm and Guarantees

| Learning Phase | Retrain Time | Unlearning Update | Unlearning Time |
|---|---|---|---|
| Co-occurrence matrix computation | $\mathcal{O}(m)$ | Updating frequencies | $\mathcal{O}(m_U)$ |
| Identify anchor words | $\mathcal{O}(n^2 + nr/\epsilon_0^2)$ | Use learned anchor words | $\mathcal{O}(1)$ |
| Recover topics from anchors | $\mathcal{O}(n^2 r + nr^2/\epsilon_0^2)$ | Projected Newton step | $\mathcal{O}(nr^2)$ |
| Head tuning $w$ (Definition 2) | ERM | Newton step | $\mathcal{O}(r^3)$ |

Table 1: Our unlearning algorithms generally have a runtime shorter than the retraining procedure. ERM denotes empirical risk minimization, and we note the training time relies on the error tolerance.

We describe our unlearning algorithm $\mathcal{U}_{\text{base}}$ to forget a set $S_f$ from a trained model (Algorithm 1), which crucially updates $C_i$ with a Newton step. We then compute $\bar{A}$ from the modified $C_i$ and apply the Gaussian mechanism to ensure indistinguishability. We describe our formal guarantee on the unlearning algorithm below, sketching out our utility preserving guarantees with respect to $A^\star$. The arguments for $R^\star$ follow analogously; we defer the discussion to the appendix.

**Theorem 2** (Utility-Preserving Unlearning on the Base Model). *Let $\mathcal{A}_{base}$ be the learning algorithm described in the prior sections and $\mathcal{U}_{base}$ be the unlearning algorithm in Algorithm 1. Then, $(\mathcal{A}_{base}, \mathcal{U}_{base})$ performs utility-preserving unlearning with deletion capacity*

$$T_{\epsilon,\delta}^{\mathcal{A}_{base}, \mathcal{U}_{base}}(m) \geq c \cdot \frac{m}{r^2 \sqrt{rn}}$$

*where $m$ is the number of training documents, $r$ is the number of topics, and $c$ is a constant dependent on $\epsilon, \delta$, and $\mathcal{D}$. The loss function $h$ used in the utility-preserving definition is the maximum entrywise error from the ground truth topic model $A^\star$.*

**Proof sketch.** The full proof can be found in Appendix B.2. We delete $m_U \leq \frac{0.001 m \epsilon_0 (\gamma p)^3}{a^2 r^2}$ points. This upper bound ensures that the anchor words are likely unchanged per Lemma 2. Recall that utility-preserving unlearning requires: (1) that the unlearned model is indistinguishable from the retrained model, and (2) that the unlearned model is not too far from the ground-truth model.

*Indistinguishability.* The Gaussian mechanism introduced in Lemma 1 allows us to make two models with a given $\ell_2$-sensitivity $(\epsilon, \delta)$-indistinguishable from each other. We bound the $\ell_2$-sensitivity of the feature extractor $A$ by noting that $\bar{A}$ is a rescaled version of $\bar{C}$.

**Lemma 4.** *For $\epsilon, \delta > 0$, the following holds for the $\bar{C}$ and the topic matrix $\bar{A}$:*

$$\|\bar{C} - C^F\|_\infty \leq c \cdot \frac{arm_U}{m\epsilon_0 \gamma p} \qquad \|\bar{A} - A^F\|_\infty \leq (ar) \cdot \|\bar{C} - C^F\|_\infty$$

Applying the Gaussian mechanism with noise $\sigma = \frac{\Delta}{\epsilon}\sqrt{2\log(1.25/\delta)}$, where $\Delta = c\sqrt{nr} \cdot \frac{(ar)^2 m_U}{m\epsilon_0 \gamma p}$ and followed by projecting the columns of $\bar{A} + \nu_A$ back to $\Delta_n$ yields the desired result.

*Utility Preservation.* We first apply Lemma 2 to show that, with high probability, the anchor words do not change when unlearning $m_U$ documents. Then, we use Lemma 8 to bound the distance between the unlearned $\bar{C}_i$ and the ground truth $C_i^\star$. Accounting for the noise added via the Gaussian mechanism completes the proof.

**Lemma 5.** *For $\epsilon, \delta > 0$, denote the unlearned model after the Gaussian mechanism described above as $\tilde{A}$. Then, $\tilde{A}$ satisfies:*

$$\mathbb{E}\left[\|\tilde{A} - A^\star\|_\infty\right] \leq c \cdot \frac{(ar)^2 m_U}{m\epsilon_0 \gamma p} \cdot \left(\sqrt{nr} \cdot \sqrt{\log(nr)} \cdot \frac{\sqrt{\log(1/\delta)}}{\epsilon} + 1\right)$$

Each of the two terms in the above equation yield a constraint on $m_U$. In particular, $m_U \leq \min\left\{\tilde{\mathcal{O}}\left(\frac{m}{r^2\sqrt{nr}}\right), \mathcal{O}\left(\frac{m}{r^2}\right)\right\}$, so setting $m_U \leq \tilde{\mathcal{O}}\left(\frac{m}{r^2\sqrt{nr}}\right)$ completes the proof.

## 5   UNLEARNING WITH RESPECT TO A DOWNSTREAM TASK

We are interested in unlearning a set of pre-training documents $S_f \subseteq S$. A topic classification task is usually defined on a subset of the topics in the dataset — for example, if the pre-training corpus contained diverse news articles, one plausible downstream task is to classify the content of a given document as containing politics or sports. Definition 1 formalizes this: a topic classification task $\mathcal{T} = (\mathbb{T}_{\text{clf}}, w^*)$ is defined on a subset of the topics $\mathbb{T}_{\text{clf}}$ and a $r$-length ground-truth labelling vector $w^* \in \mathcal{W}_{\text{head}}$, where $w^*$ only has non-zero values in positions corresponding to $\mathbb{T}_{\text{clf}}$. We describe two possible settings under which we can show utility-preserving unlearning. For the sake of exposition, we will assume for now $\tau = 0$ in downstream head tuning; the extension to inexact head tuning ($\tau > 0$), which is a more realistic regime, will be deferred to the appendix.

### 5.1   NAIVE SETTING

In the first setting, the learning algorithm $\mathcal{A}_{\text{head, naive}}$ returns the pre-trained feature extractor $A$ and the head $w$ separately. So, we must ensure that the forget set $S_f \subseteq S$ cannot be recovered from either $A$ or $w$. As such, we must necessarily perform unlearning on $A$ as described in Algorithm 1, which means that unlearning the fine-tuned model is exactly as difficult as unlearning the base model.

**Theorem 3** (Unlearning when releasing $A$ and $w$). *For a downstream task $\mathcal{T}$ with loss function $\ell_\mathcal{T}$, consider the unlearning algorithm $\mathcal{U}_{\text{head, naive}}$ that first runs Algorithm 1 to compute $\tilde{A} = \mathcal{U}_{base}(S_f, \mathcal{A}_{base}(S), T(S))$, where $(\mathcal{A}_{base}, \mathcal{U}_{base})$ performs utility-preserving unlearning (Theorem 2). Then, it fits a head $w = \arg\min_{w \in \mathcal{W}_{head}} \ell_\mathcal{T}(w; \tilde{A})$ and returns $\tilde{A}$ and $w$. We assert that $(\mathcal{A}_{\text{head, naive}}, \mathcal{U}_{\text{head, naive}})$ performs utility-preserving unlearning (Definition 4).*

---

**Algorithm 2** Unlearning algorithm for task $\mathcal{T}$ ($\mathcal{U}_{head}$)

---

**Input:** Document deletion requests $S_f \subseteq S$, statistics $T(S)$ which include $\boldsymbol{A}^S, \{\boldsymbol{C}_i^S\}_{i=1}^n, \boldsymbol{Q}^S, P$, $\mathrm{diag}(\boldsymbol{p}^S)$, $\boldsymbol{w}^S = \arg\min_{\boldsymbol{w} \in \mathcal{W}_{head}} \ell_{\mathcal{T}}(\boldsymbol{w}; \boldsymbol{A}^S)$
$\bar{\boldsymbol{A}}, \bar{\boldsymbol{R}}$ = Run Algorithm 1 ($\mathcal{U}_{base}$) up to the Gaussian mechanism
$\bar{\boldsymbol{w}} = \boldsymbol{w}^S - \boldsymbol{H}_{\boldsymbol{w}^S}^{-1} \nabla_{\boldsymbol{w}} \ell_{\mathcal{T}}(\boldsymbol{w}^S; \bar{\boldsymbol{A}})$ where $\boldsymbol{H}_{\boldsymbol{w}^S} = \nabla_{\boldsymbol{w}}^2 \ell_{\mathcal{T}}(\boldsymbol{w}^S; \bar{\boldsymbol{A}})$
**return** $(\boldsymbol{A}^S)^\dagger \bar{\boldsymbol{A}} \bar{\boldsymbol{w}} + \boldsymbol{\xi}$, in accordance with the Gaussian mechanism

---

Given the guarantee on $\tilde{\boldsymbol{A}}$ from Theorem 2, we show that this result extends to $\boldsymbol{w}$ by the well-known fact: for $\epsilon, \delta > 0$, post-processing indistinguishable quantities (Definition 3) preserves $(\epsilon, \delta)$-indistinguishability (Dwork et al., 2014). The full proof of utility preservation can be found in Appendix C, which essentially boils down to a Lipschitz condition. However, there are some downsides to this algorithm. First, it requires retraining the head $\boldsymbol{w}$ for each unlearning request, but we want to perform unlearning without access to $\mathbb{D}_{\mathrm{clf}}$. Second, repeatedly noising the base model via the Gaussian mechanism will erode its utility. We address these issues in the realistic setting.

## 5.2 REALISTIC SETTING

There is little reason to release $\boldsymbol{A}$ and $\boldsymbol{w}$ separately after fine-tuning the model, because it is unclear why one would want to use $\boldsymbol{A}$ without $\boldsymbol{w}$ or vice versa. One can obtain $\boldsymbol{A}$ directly after pre-training instead of relying on a fine-tuned model, and there is little use for $\boldsymbol{w}$ alone, because it is highly sensitive to the specific topics extracted by $\boldsymbol{A}$ and their ordering. As such, we argue for releasing the fine-tuned model as a single matrix[3] $\boldsymbol{B} = \boldsymbol{A}\boldsymbol{w}$, where $\boldsymbol{B} \in \mathbb{R}^{n \times 1}$.

**Theorem 4** (Utility-Preserving Unlearning on the Downstream Task). *Suppose that the downstream task $\mathcal{T}$ only depends on a subset of topics $\mathbb{T}_{clf} \subseteq [r]$; that is, $\boldsymbol{w}^\star = \arg\min_{\boldsymbol{v} \in \mathcal{W}_{base}} \ell_{\mathcal{T}}(\boldsymbol{v}; \boldsymbol{A}^\star)$ has non-zero entries only in the index set $\mathbb{T}_{clf}$. Denote $q := \min_{k \in \mathbb{T}_{clf}} \Pr_{\mathcal{D}}[z = k]$, and let $\mathcal{A}_{head}$ be the head tuning algorithm (Definition 2) and $\mathcal{U}_{head}$ be Algorithm 2. Then, $(\mathcal{A}_{head}, \mathcal{U}_{head})$ performs utility-preserving unlearning with deletion capacity*

$$T_{\epsilon, \delta}^{\mathcal{A}_{head}, \mathcal{U}_{head}}(m) \geq c' \cdot \frac{mq}{r\sqrt{nr}}$$

*where $c'$ is a constant dependent on $\epsilon$, $\delta$, $\mathcal{D}$, and $\mathcal{T}$.*

The full proof is in Appendix C, including the worst case of $\mathbb{T}_{\mathrm{clf}} = [r]$. When the task relies heavily on every single topic (i.e., $q = 1/ar$), the above guarantee is equivalent to the one in the pre-training phase. However, in most realistic settings, the downstream task will only depend on a subset of the latent topics in the corpus. In this case, $q > 1/ar$, and we can unlearn more points without degrading the utility of the model. Intuitively this makes sense too; the more reliance $\mathcal{T}$ has on a rare topic, the less adversarial deletion it can tolerate.

**Proof sketch.** We again assume that we are deleting $m_U \leq \frac{0.001 m \epsilon_0 (\gamma p)^3}{a^2 r^2}$ points. For any modification made to $\boldsymbol{A}$, there is an equivalent modification that can be made to $\boldsymbol{w}$ instead such that $\boldsymbol{B} = \boldsymbol{A}\boldsymbol{w}$ is preserved, so we do not need to update $\boldsymbol{A}$. We look for $\boldsymbol{v} \in \mathcal{W}_{\mathrm{head}}$ such that $\boldsymbol{A}^S \boldsymbol{v} = \boldsymbol{A}^F \boldsymbol{w}^F$, where $\boldsymbol{w}^F$ is the head learned on $\boldsymbol{A}^F$. It can be shown that $\bar{\boldsymbol{A}}^S$ has a unique pseudoinverse since it is full rank; naturally, we set $\boldsymbol{v} = \boldsymbol{A}^{S\dagger} \boldsymbol{A}^F \boldsymbol{w}^F$, thereby ensuring privacy even if one recovers a part of $\boldsymbol{A}$ from $\boldsymbol{B} = \boldsymbol{A}\boldsymbol{w}$. We furthermore define $\bar{\boldsymbol{v}}$ that is fit to the unlearned model before the Gaussian mechanism, $\bar{\boldsymbol{v}} = \boldsymbol{A}^{S\dagger} \bar{\boldsymbol{A}} \bar{\boldsymbol{w}}$. We now need to show $\boldsymbol{v}$ and $\bar{\boldsymbol{v}}$ satisfy both the indistinguishability and utility preservation conditions in Definition 4.

*Indistinguishability.* Let $\bar{\boldsymbol{w}}^\star = \arg\min_{\boldsymbol{v} \in \mathcal{W}_{\mathrm{head}}} \ell_{\mathcal{T}}(\boldsymbol{v}; \bar{\boldsymbol{A}})$ denote the result of head tuning $\bar{\boldsymbol{A}}$, and let $\bar{\boldsymbol{w}}$ be the result of taking a Newton step on $\boldsymbol{w}$ (see Algorithm 2). Then by triangle inequality,

$$\|\bar{\boldsymbol{A}}\bar{\boldsymbol{w}} - \boldsymbol{A}^F \boldsymbol{w}^F\|_2 \leq \|\bar{\boldsymbol{A}}\bar{\boldsymbol{w}} - \bar{\boldsymbol{A}}\bar{\boldsymbol{w}}^\star\|_2 + \|\bar{\boldsymbol{A}}\bar{\boldsymbol{w}}^\star - \boldsymbol{A}^F \bar{\boldsymbol{w}}^\star\|_2 + \|\boldsymbol{A}^F \bar{\boldsymbol{w}}^\star - \boldsymbol{A}^F \boldsymbol{w}^F\|_2$$

---

[3]One can generalize this to the case where the downstream task is a $C$-way classification, in which case $\boldsymbol{B} \in \mathbb{R}^{n \times C}$.

Informally, the first term is controlled by the error in the Newton step approximation, and the third term is bounded by the error to the retrained $\boldsymbol{w}^F$. The remaining term can be rewritten as $\|(\bar{\boldsymbol{A}} - \boldsymbol{A}^F)(\bar{\boldsymbol{w}}^\star - \boldsymbol{w}^\star) + (\bar{\boldsymbol{A}} - \boldsymbol{A}^F)\boldsymbol{w}^\star\|$, where the first term can be bounded using the same technique use to prove Lemmas 4 and 5. The second term can be bounded by noting that $\boldsymbol{w}^\star$ is sparse, which yields the below lemma that plays a crucial role in establishing the improved deletion capacity.

**Lemma 6** (Modification of Lemma 4 for downstream task). *For $\epsilon, \delta > 0$,*

$$\|\bar{\boldsymbol{A}} - \boldsymbol{A}^F\|_\infty \leq \frac{1}{q} \cdot \|\bar{\boldsymbol{C}} - \boldsymbol{C}^F\|_\infty = c \cdot \frac{1}{q} \cdot \frac{arm_U}{m\epsilon_0\gamma p}$$

As in the pre-training case, we can now set the noise scale in the Gaussian mechanism and complete the proof. In the worst case, when the downstream task depends on *every* topic, then $q = 1/ar$, and we recover Lemma 4; however, this is unlikely to happen in practice.

*Utility Preservation.* We compare the value of $\boldsymbol{v}$ after the Gaussian mechanism $\tilde{\boldsymbol{v}} = \bar{\boldsymbol{v}} + \nu_{\bar{\boldsymbol{v}}}$ to what it would be for the ground-truth model $\boldsymbol{v}^\star = (\boldsymbol{A}^S)^\dagger \boldsymbol{A}^\star \boldsymbol{w}^\star$. We again rely the sparsity of $\boldsymbol{w}^\star$ and bound $\mathbb{E}[\|\bar{\boldsymbol{v}} - \boldsymbol{v}^\star\|_\infty]$ in Lemma 31.

## 6 RELATED WORKS

**Provable unlearning.** One ideally wants the unlearned model to behave identically to one that was retrained from scratch with the forget set removed from the training data (Cao & Yang, 2015; Bourtoule et al., 2021; Gupta et al., 2021). This is difficult to achieve in many settings, so there are several notions of approximate unlearning (Ginart et al., 2019; Guo et al., 2020; Neel et al., 2021) reminiscent of differential privacy (Dwork et al., 2014). Most relevant to our work is the notion of $(\epsilon, \delta)$-unlearning introduced in Sekhari et al. (2021), which we adapt to construct Definition 4. Our work focuses on deriving unlearning guarantees in the pre-training and fine-tuning pipeline. Golatkar et al. (2020) is closest to our work. They show considerably weaker guarantees on unlearning information with respect to probes fit to the weights. In contrast, our work is focused on realistic topic classification tasks and demonstrates strong guarantees (Definition 4). Recent works have extended notions of certified unlearning to nonconvex settings. Zhang et al. (2024a); Mu & Klabjan (2024); Chien et al. (2024) provide unlearning algorithms without deletion capacity guarantees. Qiao et al. (2024) also proposes an unlearning method for non-convex settings but analyzes its deletion capacity in a convex setting. Our work extends beyond the convex setting to provide provable unlearning methods and corresponding deletion capacity analysis for non-convex models.

**Theoretical analysis of pre-training and fine-tuning.** Our downstream task definition (Section 2.2) is inspired by works on transfer learning in language models (Saunshi et al., 2021; Wei et al., 2021; Wu et al., 2023; Kumar et al., 2022), contrastive learning (Lee et al., 2021; HaoChen & Ma, 2023), and meta-learning (Chua et al., 2021; Collins et al., 2022; Yüksel et al., 2024).

## 7 CONCLUSION

This work uses topic models to develop the first provable guarantees on unlearning in the modern-day pre-training and fine-tuning paradigm. We propose two unlearning algorithms that can effectively and efficiently unlearn from both the pre-trained model (Algorithm 1 and Theorem 2) and the fine-tuned model (Algorithm 2 and Theorem 4). Notably, we find that it is easier, in terms of the deletion capacity (Definition 4), to unlearn pre-training data from the fine-tuned model, and we can do so without modifying the pre-trained base model. Our findings suggest that task-specific unlearning is easier than full model unlearning, providing a promising path forward to design efficient algorithms for large-scale models.

The most notable limitation of our work is that our usage of topic models, which permit a tractable analysis but cannot capture interesting features of modern-day language models (e.g., their autoregressive nature). Moreover, with the growing popularity of foundation models, there is scholarly discussion around meaningful definitions of unlearning and how they can be measured (Thudi et al., 2022; Lee et al., 2024). Our work focuses on traditional notions of unlearning centered on differential privacy (see Definition 4), but we hope to extend these definitions to capture additional features of generative models that are salient to their real-world uses.

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

# A  PRECISE DESCRIPTION OF $\mathcal{A}_{\text{BASE}}$

## A.1  COMPLETE DESCRIPTION

---

**Algorithm 3** High level learning algorithm ($\mathcal{A}$)

---

**Input:** document corpus $S = \{d_i\}_{i=1}^m$, anchor word tolerance $\epsilon_0$
**Output:** matrices $A$,$R$
$Q$ = word co-occurrences
$\bar{Q}$ = row-normalized $Q$
$P$ = RecoverAnchors($\{\bar{Q}_1, \ldots, \bar{Q}_n\}$)
$A, R$ = RecoverTopics($Q, S$)
**return** $A, R$

---

---

**Algorithm 4** RecoverAnchors, same as Arora et al. (2012a)

---

**Input:** Row-normalized co-ocurrence matrix $\bar{Q}$ and $\epsilon_0$ tolerance parameter
**Output:** $r$ points of this perturbed simplex close to the vertices of the actual simplex
Project the rows to a randomly chosen $4 \log n / \epsilon_0^2$ dimensional subspace
$S \leftarrow \{\bar{Q}_i\}$ where $\bar{Q}_i$ is the furthest point from the origin
**for** $i$ in $1, \ldots, r-1$ **do**
    Let $\bar{Q}_j$ be the row of $Q$ with largest distance to span($S$)
    $S \leftarrow S \cup \{\bar{Q}_j\}$
**end for** $S = \{\bar{Q}_{s_1}, \ldots, \bar{Q}_{s_r}\}$
**for** $i$ in $1, \ldots, r$ **do**
    Let $\bar{Q}_j$ be the point that has largest distance to span($S \setminus \{\bar{Q}_{s_i}\}$)
    Remove $\bar{Q}_{s_i}$ from $S$ and insert $\bar{Q}_j$ into $S$
**end for**
**return** S

---

---

**Algorithm 5** Recover Topics, from Arora et al. (2012a)

---

**Input:** Co-ocurrence matrix $Q$, anchor words $P = \{s_1, \ldots, s_k\}$, tolerance parameter $\epsilon_0$
**Output:** Matrices $A, R$
$\bar{Q}$ = row normalized $Q$
Store the normalization constants $p = Q\mathbf{1}$
**for** $i$ in $1, \ldots, n$ **do**
    Solve $C_i = \underset{v \in \Delta_r}{\arg\min} \|\bar{Q}_i - v^\top \bar{Q}_P\|^2$
    up to $\epsilon_0$ accuracy
**end for**
$A' = \text{diag}(p)C$
$A$ = column-sum-one normalized $A'$
$R = A^\dagger Q A^{\dagger\top}$ where $A^\dagger$ is the pseudoinverse of $A$
**return** $A, R$

---

More formally, the co-occurrence matrix is constructed as follows. For each document, let $H_d \in \mathbb{R}^n$ be the frequency vector of each word in the document; the sum of its entries should be $L$. Then, for a document $d$, consider the matrix

$$G_d := \tilde{H}_d \tilde{H}_d^\top - \hat{H}_d$$

where

$$\tilde{H}_d := \frac{H_d}{\sqrt{L(L-1)}}$$

$$\hat{H}_d := \frac{\text{diag}(H_d)}{L(L-1)}$$

In particular, the denominator term $L(L-1)$ is precisely the number of co-occurences in each document, by simple combinatorics, and it can be seen that the sum of the entries of $\boldsymbol{G}_d$ is always 1. Our co-ocurrence matrix $\boldsymbol{Q}$ is defined to be

$$\boldsymbol{Q} := \frac{1}{m} \sum_{i=1}^{m} \boldsymbol{G}_d$$

so that $\boldsymbol{Q}$ also has entries that sum to 1. By linearity of expectation, we have

$$\mathbb{E}[\boldsymbol{Q}] = \mathbb{E}[\boldsymbol{G}_d] = \boldsymbol{A}^\star \mathbb{E}[\boldsymbol{X}_d \boldsymbol{X}_d^\top] \boldsymbol{A}^{\star\top}$$

which implies that as the number of documents increases, $\boldsymbol{Q}$ concentrates around $\boldsymbol{A}\mathbb{E}[\boldsymbol{X}\boldsymbol{X}^\top]\boldsymbol{A}^\top = \mathbb{E}[\boldsymbol{M}\boldsymbol{M}^\top]$. Therefore, we should expect $\boldsymbol{A}^\dagger \boldsymbol{Q} \boldsymbol{A}^{\dagger\top}$ to concentrate around $\mathbb{E}[\boldsymbol{X}\boldsymbol{X}^\top] = \boldsymbol{R}^\star$.

### A.2 SKETCH: POPULATION ANALYSIS

To understand this algorithm, consider the setting where we have infinitely many documents. Specifically, consider two words $w_1, w_2$ in a document and their respective topics $z_1, z_2$. Then, this population co-occurrence matrix $\boldsymbol{Q}$ will have elements $\boldsymbol{Q}_{i,j} = \Pr[w_1 = i, w_2 = j]$, and the row-normalized co-occurrence matrix $\bar{\boldsymbol{Q}}$ will have entries $\bar{\boldsymbol{Q}}_{i,j} = \Pr[w_2 = j | w_1 = i]$. Moreover, we have that $\boldsymbol{A}_{i,k} = \Pr[w_1 = i | z_1 = k] = \Pr[w_2 = i | z_2 = k]$.

Consider the set of anchor words $P = \{s_1, \ldots, s_r\} \subseteq [n]$, where $s_k$ is the anchor word for topic $k$. Then, observe that for an anchor word row $s_k$ of $\bar{\boldsymbol{Q}}$, it holds that

$$\begin{aligned}
\bar{\boldsymbol{Q}}_{s_k,j} = \Pr[w_2 = j | w_1 = s_k] &= \sum_{k'} \Pr[z_1 = k' | w_1 = s_k] \Pr[w_2 = j | w_1 = s_k, z_1 = k'] \\
&= \Pr[w_2 = j | w_1 = s_k, z_1 = k] \\
&= \Pr[w_2 = j | z_1 = k]
\end{aligned}$$

where the second line follows from only $\Pr[z_1 = k | w_1 = s_k] = 1$ in the summation, and the last line follows from $w_2, w_1$ are conditionally independent given $z_1$. Furthermore, for non-anchor word rows $i$ of $\bar{\boldsymbol{Q}}$, it holds that

$$\bar{\boldsymbol{Q}}_{i,j} = \sum_{k} \Pr[z_1 = k | w_1 = i] \Pr[w_2 = j | z_1 = k]$$

where again we use that $w_2, w_1$ are conditionally independent $z_1$. For a word $i$, let $\boldsymbol{C}_i \in \mathbb{R}^r$ be the vector such that $\boldsymbol{C}_{i,k} := \Pr[z_1 = k | w_1 = i]$. Then, it holds that $\bar{\boldsymbol{Q}}_i = \boldsymbol{c}_i^\top \bar{\boldsymbol{Q}}_S$, where $\bar{\boldsymbol{Q}}_S$ is the submatrix of $\bar{\boldsymbol{Q}}$ constrained to the anchor word rows. In other words, for every word $i$, $\bar{\boldsymbol{Q}}_i$ is a convex combination of rows of $\bar{\boldsymbol{Q}}_S$.

In the algorithm, one can see that $\boldsymbol{A}'_{i,k} = \boldsymbol{C}_{i,k} \boldsymbol{p}_i$. Normalizing this along each column, we obtain

$$\boldsymbol{A}_{i,k} = \frac{\boldsymbol{C}_{i,k} \boldsymbol{p}_i}{\sum_{i'} \boldsymbol{C}_{i',k} \boldsymbol{p}_{i'}} = \frac{\Pr[z_1 = k | w_1 = i] \Pr[w_1 = i]}{\sum_{i'} \Pr[z_1 = k | w_1 = i'] \Pr[w_1 = i']} = \Pr[w_1 = i | z_1 = k]$$

Hence, in the infinite document limit, this algorithm recovers the ground truth $\boldsymbol{A}^\star, \boldsymbol{R}^\star$.

## B FROM PROPERTIES OF THE LEARNING ALGORITHM TO THE PROOF OF THEOREM 2

We first give the formal statement of Theorem 2.

**Theorem 5** (Formal statement of Theorem 2). *Let $\mathcal{A}_{base}$ be the learning algorithm described in the prior sections and $\mathcal{U}_{base}$ be the unlearning algorithm in Algorithm 1. Then, $(\mathcal{A}_{base}, \mathcal{U}_{base})$ performs utility-preserving unlearning with deletion capacity*

$$T_{\epsilon,\delta}^{\mathcal{A}_{base}, \mathcal{U}_{base}}(m) \geq c \cdot \min \left\{ \frac{m\epsilon}{r^2 \sqrt{rn \log 1/\delta}}, \frac{0.001m}{r^2} \right\}$$

*where $m$ is the number of training documents, $r$ is the number of topics, and $c$ is a constant dependent on $\mathcal{D}$. The loss function $h$ used in the utility-preserving definition is the maximum entrywise error from the ground truth topic model $\boldsymbol{A}^\star$.*

### B.1 Preliminaries

When the norm is not specified, we assume that it is the Euclidean norm $\|\cdot\|_2$. We now start off with a technical assumption on the precision of the learning algorithm.

**Assumption 2.** $\epsilon_0 \leq O(1/\sqrt{nr})$.

**Assumption 3.** *Every word appears with probability $\epsilon_0/4ar$ without loss of generality; see discussion in Arora et al. (2012b). Essentially, less probable words can be combined in a sense to form a single category of "rare" words.*

We recall the definitions from Arora et al. (2012a).

**Definition 6** ($\beta$-robust simplex). A simplex $P$ is $\beta$-robust if for every vertex $v$ of $P$, the $\ell_2$ distance between $v$ and the convex hull of the rest of the vertices as at least $\beta$.

**Definition 7.** Let $\{a_i\}_{i=1}^n$ be a set of points whose convex hull is a simplex with vertices $\{v_i\}_{i=1}^r$. We say a set of $r$ points is $\epsilon$-close the vertex set $\{v_i\}_{i=1}^r$ if each of the $r$ points is $\epsilon$-close in $\ell_2$ distance to a different vertex in this vertex set.

The following result will be used throughout our proof.

**Proposition 1** (Arora et al. (2012b)). $\bar{Q}_P^\star$ *in population is $\gamma p$-robust.*

We now list the high probability events we condition on throughout our proof. These follow from previous results in Arora et al. (2012a); they concern the properties of the output of the learning algorithm.

**Proposition 2.** *With high probability, in our regime of $m$, the following hold:*

- *The correct anchor words are selected.*

- *Each word appears at least $O\left(\frac{m\epsilon_0}{4ar}\right)$ times.*

- *The error in the empirical matrix $\hat{Q}$ is entrywise at most $\tilde{O}(1/\sqrt{m})$ from the population $Q^\star$.*

We also utilize the following two key lemmas from Arora et al. (2012a) that we touched upon in the main paper.

**Lemma 7** (Approximation Guarantee on Anchor Words). *Suppose each row of $\bar{Q}$ is at most $\delta$ distance away from the ground truth $\gamma p$-robust simplex $\bar{Q}^\star$ in $\ell_2$ norm. If $20r\delta/(\gamma p)^2 < \gamma p$, then the set of anchor words found by the algorithm is $O(\delta/\gamma p)$-close to the ground truth anchor words.*

**Lemma 8.** *When $20r\delta/(\gamma p)^2 < \gamma p$, it holds for every word $i$ that $C_i$ has entrywise error $O(\delta/(\gamma p)^2)$ from $C_i^\star$.*

### B.2 Proof of Theorem 2

The following are lemmas bounding the relation between $\bar{Q}_i^S, \bar{Q}_i^F, \bar{Q}_i^\star$.

**Lemma 9.** *After training, the error of each row of $\bar{Q}^S$ is at most $\delta_2 := O\left(\sqrt{\frac{4ar}{m\epsilon_0}}\right)$. That is, $\|\bar{Q}_i^S - \bar{Q}_i^\star\| \leq \delta_2$ for all words $i$.*

*Importantly, note that*

$$20r\delta_2/(\gamma p)^2 < \gamma p$$

*This implies that the anchor words of $\bar{Q}_i^S$ are $O(\delta_2/(\gamma p))$ close to the anchor words of $\bar{Q}_i^\star$.*

*Consequently, it holds that*

$$\|C^S - C^\star\|_\infty \leq O(\delta_2/(\gamma p)^2)$$

*Proof.* The first part follows directly from the fact that if the number of documents $m = \tilde{\Omega}(1/\epsilon_Q^2)$, then $\|\bar{Q}_i^S - \bar{Q}_i^\star\| \leq \delta_2$ for each row $i$. To show that

$$20r\delta_2/(\gamma p)^2 < \gamma p$$

we note that by the sample complexity guarantee,

$$m\epsilon_0 \geq \tilde{O}\left(\frac{ar^3}{(\gamma p)^6}\right)$$

which implies that

$$\delta_2 \leq \tilde{O}\left(\frac{(\gamma p)^3}{r}\right)$$

as desired. □

**Lemma 10.** *When we delete $m_U \leq \frac{0.001 m\epsilon_0(\gamma p)^3}{a^2 r^2}$, it holds that*

$$\|\bar{\boldsymbol{Q}}_i^F - \bar{\boldsymbol{Q}}_i^S\| \leq \frac{m_U}{m\epsilon_0/4ar} = \frac{4ar m_U}{m\epsilon_0}$$

*In particular, this is smaller than*

$$\frac{0.001 m\epsilon_0(\gamma p)^3}{a^2 r^2} \cdot \frac{1}{m\epsilon_0/4ar} = \frac{0.004(\gamma p)^3}{ar}$$

*Proof.* For a word $i$, consider the change in $\bar{Q}_i$ after deletion requests. Let $F$ be the initial sum of the the $i$th row of $\boldsymbol{Q}$. Each coordinate $j \in [n]$ will change as follows:

$$\delta_j = \frac{f_j - t_j}{F - m_U} - \frac{f_j}{F} = \frac{m_U f_j - F t_j}{F(F - m_U)}$$

where $f_j$ is the initial number of coocurrences of words $i, j$ and $t_j$ is the number of documents removed that have this cooccurrence. Moreover, $F$ is the number of initial occurrences of word $i$, and $T$ is the number of deletions of the word $i$. From the previous lemma, it holds that $F \geq m\epsilon_0/4ar$, and that $m_U \geq \sum_{j=1}^n t_j$ Hence, it follows that the squared Euclidean norm of the change is:

$$\sum_{j=1}^n \delta_j^2 = \frac{1}{F^2(F-T)^2} \sum_{j=1}^n (m_U f_j - F t_j)^2 \leq \frac{2F^2 m_U^2}{F^2(F-m_U)^2} \leq 2\left(\frac{m_U}{F - m_U}\right)^2$$

Hence, for the regime where $m_U \leq \frac{0.001 m\epsilon_0(\gamma p)^3}{a^2 r^2}$, we have

$$\|\bar{\boldsymbol{Q}}_i^S - \bar{\boldsymbol{Q}}_i^F\| \leq \sqrt{2}\frac{m_U}{F - m_U} \lesssim \frac{m_U}{F} \lesssim \frac{4ar m_U}{m\epsilon_0}$$

Of particular notice is that when $m_U$ is taken as large as possible, this is at most

$$\frac{0.001 m\epsilon_0(\gamma p)^3/a^2 r^2}{m\epsilon_0/4ar} = 0.004(\gamma p)^3/ar$$

□

We now combine the above two with triangle inequality.

**Lemma 11.** *Hence, it holds that*

$$\|\bar{\boldsymbol{Q}}_i^F - \bar{\boldsymbol{Q}}_i^\star\| \leq \frac{4ar m_U}{m\epsilon_0} + \delta_2 = \frac{4ar m_U}{m\epsilon_0} + O\left(\sqrt{\frac{4ar}{m\epsilon_0}}\right) =: \delta_2'$$

*Importantly, note that*

$$20r\delta_2'/(\gamma p)^2 < \gamma p$$

*This implies that the anchor words of $\bar{\boldsymbol{Q}}_i^F$ are $O(\delta_2'/(\gamma p))$ close to the anchor words of $\bar{Q}_i^\star$.*

*Consequently, it holds that*

$$\|\boldsymbol{C}^F - \boldsymbol{C}^\star\|_\infty \leq O(\delta_2'/(\gamma p)^2)$$

*Proof.* The first part follows from triangle inequality, and the second part follows from lemma B.1 from Arora et al. (2012a). $\qquad\square$

We now bound what happens to $\|C^F - C^S\|_\infty$. First, we have that the perturbed simplex $\bar{Q}_P^S$ is $\gamma p/2$-robust.

**Lemma 12.** *The perturbed simplex $\bar{Q}_P^S$ is $\gamma p/2$-robust.*

*Proof.* This is because of Lemma A.1 in Arora et al. (2012a). Since $10\sqrt{r}\delta_2 < \gamma p$, the result of that lemma applies. $\qquad\square$

Hence, we will apply Lemma B.1 from Arora et al. (2012a) on $C^S$ to say something about $\|C^F - C^S\|_\infty$.

**Lemma 13.** *Recall that when we delete $m_U \leq \frac{0.001m\epsilon_0(\gamma p)^3}{a^2r^2}$, it holds that*

$$\|\bar{Q}_i^F - \bar{Q}_i^S\| \leq \frac{m_U}{m\epsilon_0/4ar} = \frac{4arm_U}{m\epsilon_0}$$

*Importantly, note that*

$$20r\left(\frac{4arm_U}{m\epsilon_0}\right)/(\gamma p/2)^2 < \gamma p/2$$

*This implies that the anchor words of $\bar{Q}_i^F$ are $\frac{4arm_U/m\epsilon_0}{\gamma p/2}$ close to the anchor words of $\bar{Q}_i^S$. By lemma B.1 from Arora et al. (2012a), it holds that*

$$\|C^F - C^S\|_\infty \leq O\left(\frac{4arm_U}{m\epsilon_0}/(\gamma p/2)^2\right)$$

*Observe that this is smaller than $O((\gamma p)/ar)$.*

We now deal with the Hessian step that we had took to prevent retraining the $C_i$'s. In particular, we will denote $\bar{C}$ to be our estimated new $C$.

First, a lemma to say that our Hessian step is full rank and has a lower bound on its minimum singular value.

**Lemma 14.** *When we delete $m_U \leq \frac{0.001m\epsilon_0(\gamma p)^3}{a^2r^2}$ samples, it holds that the minimum eigenvalue of $\bar{Q}_P^F\bar{Q}_P^F$ is at least $\gamma p/2$.*

*Proof.* Follows from Lemma A.3 in Arora et al. (2012a). $\qquad\square$

**Lemma 15.** *When we delete $m_U \leq \frac{0.001m\epsilon_0(\gamma p)^3}{a^2r^2}$ samples, it holds for all $i$,*

$$\|C_i^F - \bar{C}_i^F\| \leq \frac{4}{\gamma p}\left(\delta_2 + \frac{4arm_U}{m\epsilon_0}\right)$$

*Proof.* For the case of $d(\cdot, \cdot)$ being the squared loss, we will denote the following:

$$C_{i,\text{uncon}} := \arg\min_C \|\bar{Q}_P^{F\top}C - \bar{Q}_i^{F\top}\|^2 = (\bar{Q}_P^F\bar{Q}_P^{F\top})^{-1}\bar{Q}_P^F\bar{Q}_i^{F\top}$$
$$\bar{C}_i^F := \text{proj}_{\Delta_r}(C_{i,\text{uncon}})$$
$$C_i^F := \arg\min_{C\in\Delta_r} \|\bar{Q}_P^{F\top}C - \bar{Q}_i^{F\top}\|^2$$

In particular, the Newton step plus projection outputs $C_{i,\text{proj}}$. First, observe that by one of the anchor word lemmas,

$$\min_C \|\bar{Q}_P^{F\top}C - \bar{Q}_i^{F\top}\| = \|\bar{Q}_P^{F\top}C_{i,\text{uncon}} - \bar{Q}_i^{F\top}\| \leq \|\bar{Q}_P^{F\top}C_i^F - \bar{Q}_i^{F\top}\| \leq \delta_2 + \frac{4arm_U}{m\epsilon_0}$$

The last inequality follows from the fact that $\bar{Q}_P^F$ is a perturbed version of $\bar{Q}_P^S$, and $\bar{Q}_P^S$ is a perturbed version of $\bar{Q}_P^\star$. Hence, we will bound

$$
\begin{aligned}
\|\bar{C}_i^F - C_i^F\| &= \|\text{proj}_{\Delta_r}(C_{i,\text{uncon}}) - \text{proj}_{\Delta_r}(C_i^F)\| \\
&\leq \|C_{i,\text{uncon}} - C_i^F\| \\
&\leq \frac{1}{\sigma_{\min}} \|\bar{Q}_P^{F\top}(C_{i,\text{uncon}} - C_i^F)\| \\
&\leq \frac{1}{\sigma_{\min}} (\|\bar{Q}_i^{F\top} - \bar{Q}_P^{F\top} C_i^F\| + \|\bar{Q}_P^{F\top} C_{i,\text{uncon}} - \bar{Q}_i^{F\top}\|) \\
&\leq \frac{2}{\sigma_{\min}} \left(\delta_2 + \frac{4arm_U}{m\epsilon_0}\right)
\end{aligned}
$$

where $\sigma_{\min}$ is the smallest singular value of $\bar{Q}_i^{F\top}$, which is guaranteed to be full rank per the previous lemma. Due to a result in Arora et al. (2012a), this $\sigma_{\min} \geq (\gamma p)/2$. This gives us that the whole quantity is at most

$$
\frac{4}{\gamma p}\left(\delta_2 + \frac{4arm_U}{m\epsilon_0}\right)
$$

$\square$

**Corollary 1.** *We have that*

$$
\|C^F - \bar{C}^F\|_\infty \leq \frac{4}{\gamma p}\left(\delta_2 + \frac{4arm_U}{m\epsilon_0}\right)
$$

*since the $\ell_\infty$ norm is upper bounded by the $\ell_2$ norm.*

**Lemma 16.** *The following are true.*

- $\|C^F - \bar{C}^F\|_\infty \leq \frac{4}{\gamma p}\left(\delta_2 + \frac{4arm_U}{m\epsilon_0}\right)$

- $\|\bar{C}^F - C^\star\|_\infty \leq \|\bar{C}^F - C^F\|_\infty + \|C^F - C^\star\|_\infty \leq \frac{4}{\gamma p}\left(\delta_2 + \frac{4arm_U}{m\epsilon_0}\right) + O(\delta_2'/(\gamma p)^2)$

From this, we can bound the errors on the topic matrix.

**Lemma 17.** *The following are true.*

- $\|A^F - \bar{A}\|_\infty \leq O(ar\|C^F - \bar{C}^F\|_\infty)$

- $\|\bar{A} - A^\star\|_\infty \leq O(ar\|\bar{C}^F - C^\star\|_\infty)$

- $\|A^S - A^F\|_\infty \leq O(ar\|C^F - C^S\|_\infty)$

*Proof.* Note that entries $A_{i,k}$ are

$$
A_{i,k} = \frac{C_{i,k} \Pr[w = i]}{\Pr[z = k]}
$$

Therefore, the perturbation in $A$ will be the perturbation in $C$ multiplied by $ar$, since the denominator is lower bounded by $1/ar$ due to the topic imbalance constant. $\square$

Now, we give a new lemma.

**Proposition 3.** *When $m_U \geq \Omega(\sqrt{\frac{m\epsilon_0}{4ar}})$, we have that*

$$
\delta_2' = \delta_2 + \frac{4arm_U}{m\epsilon_0} = \sqrt{\frac{4ar}{m\epsilon_0}} + \frac{4arm_U}{m\epsilon_0} \leq O\left(\frac{arm_U}{m\epsilon_0}\right)
$$

Now, we analyze what happens given that $\Omega\left(\sqrt{\frac{m\epsilon_0}{4ar}}\right) \leq m_U \leq \frac{0.001 m\epsilon_0 (\gamma p)^3}{a^2 r^2}$.

**Lemma 18.** *For $\epsilon, \delta > 0$, the deletion capacity satisfies*

$$T_{\epsilon,\delta}^{\mathcal{A},\mathcal{U}}(m) \geq \tilde{\Omega}\left(\frac{m}{r^2\sqrt{nr}}\right)$$

*Proof.* Recall that

$$\|\bar{\boldsymbol{A}} - \boldsymbol{A}^\star\|_\infty \leq O(ar\delta_2'(1/\gamma p + 1/(\gamma p)^2)) \leq O\left(\frac{(ar)^2 m_U}{m\epsilon_0 \gamma p}\right)$$

Moreover, we also have that

$$
\begin{aligned}
\|\bar{\boldsymbol{A}} - \boldsymbol{A}^F\|_\infty &\leq O(ar\|\boldsymbol{C}^F - \bar{\boldsymbol{C}}^F\|_\infty) \\
&\leq O\left(\frac{4ar\delta_2'}{\gamma p}\right) \\
&\leq O\left(\frac{(ar)^2 m_U}{m\epsilon_0 \gamma p}\right)
\end{aligned}
$$

Note that $\boldsymbol{A}$ has $\ell_2$ sensitivity $O\left(\sqrt{nr}\frac{(ar)^2 m_U}{m\epsilon_0 \gamma p}\right)$. We now apply the Gaussian mechanism to the matrix $A$ entrywise with noise

$$\sigma = \frac{O\left(\sqrt{nr}\frac{(ar)^2 m_U}{m\epsilon_0 \gamma p}\right)}{\epsilon}\sqrt{2\log(1.25/\delta)}$$

From this, we obtain that

$$
\begin{aligned}
\mathbb{E}\left[\|\tilde{\boldsymbol{A}} - \boldsymbol{A}^\star\|_\infty\right] &\leq \mathbb{E}\left[\max_{i,k}|\nu_{i,k}|\right] + \mathbb{E}\left[\|\bar{\boldsymbol{A}} - \boldsymbol{A}^\star\|_\infty\right] \\
&\leq O\left(\sqrt{nr}\cdot\frac{(ar)^2 m_U}{m\epsilon_0 \gamma p}\cdot\sqrt{\log(nr)}\cdot\frac{\sqrt{\log(1/\delta)}}{\epsilon}\right) + O\left(\frac{(ar)^2 m_U}{m\epsilon_0 \gamma p}\right)
\end{aligned}
$$

Finally, this says that when

$$m_U \leq \tilde{\Omega}\left(\frac{m}{r^2\sqrt{nr}}\right)$$

we have that the utility is preserved up to constant amount, say 0.01. $\square$

This proves Theorem 2. It is straightforward to continue the perturbation analysis for the topic-topic covariance matrix $\boldsymbol{R}^\star$ and prove similar deletion capacity rates.

## C  DOWNSTREAM TASK PROOFS

Recall the algorithm for learning the downstream task head.

---

**Algorithm 6** Learning algorithm for task $\mathcal{T}$ ($\mathcal{A}_{head}$)

---

**Input:** document corpus $S = \{d_i\}_{i=1}^m$, anchor word tolerance $\epsilon_0$
$\boldsymbol{A}, \boldsymbol{R} = \mathcal{A}_{base}(S)$
**return** $\arg\min_{\boldsymbol{w}\in\mathcal{W}_{head}} \ell_\mathcal{T}(\boldsymbol{w}; \boldsymbol{A})$

---

**Assumption 4.** *For any $A$, $\ell_\mathcal{T}$ is $\lambda$-strongly convex with respect to $w$.*

Since our topic matrix $A$, can only take on a bounded support (i.e. the set of matrices where each row is on the probability simplex), it is natural to say that the set of values $w^\star(A)$ takes on over all topic matrices $A$ is bounded in a certain sense. As such, we also assume the following:

**Assumption 5.** *For any base model $\boldsymbol{A}$, the vector $\boldsymbol{v}$ such that $\boldsymbol{v} = \arg\min_{\boldsymbol{w}} \ell_{\mathcal{T}}(\boldsymbol{w}; \boldsymbol{A})$ satisfies $\|\boldsymbol{v}\|_2 \leq B$.*

**Assumption 6.** *For any $\boldsymbol{A}$, $\ell_{\mathcal{T}}$ is L-Lipschitz with respect to $\boldsymbol{w}$ and the $\ell_2$ norm, and is $L_2$-Hessian Lipschitz with respect to $\boldsymbol{w}$ and the $\ell_2$ norm. In other words,*

$$\|\ell_{\mathcal{T}}(\boldsymbol{A}, \boldsymbol{w}_1) - \ell_{\mathcal{T}}(\boldsymbol{A}, \boldsymbol{w}_2)\|_2 \leq L\|\boldsymbol{w}_1 - \boldsymbol{w}_2\|_2$$

$$\|\nabla_{\boldsymbol{w}}^2 \ell_{\mathcal{T}}(\boldsymbol{A}, \boldsymbol{w}_1) - \nabla_{\boldsymbol{w}}^2 \ell_{\mathcal{T}}(\boldsymbol{A}, \boldsymbol{w}_2)\|_2 \leq L_2\|\boldsymbol{w}_1 - \boldsymbol{w}_2\|_2$$

**Assumption 7.** *For any $w$, $\nabla_w \ell_{\mathcal{T}}$ is $L_\infty$-Lipschitz with respect to $A$ and the $\ell_\infty$ norm; that is,*

$$\|\nabla_{\boldsymbol{w}} \ell_{\mathcal{T}}(\boldsymbol{A}, \boldsymbol{w}) - \nabla_{\boldsymbol{w}} \ell_{\mathcal{T}}(\tilde{\boldsymbol{A}}, \boldsymbol{w})\|_2 \leq L_\infty \|\boldsymbol{A} - \tilde{\boldsymbol{A}}\|_\infty$$

We give a helper lemma that $(\epsilon, \delta)$-indistinguishability is immune to post processing.

**Lemma 19** (Post-processing immunity). *Consider two random variables $\theta_1, \theta_2 \in \Theta$ that are $(\epsilon, \delta)$-indistinguishable. Then, for any arbitrary mapping $f : \Theta \to \Theta'$, it holds that $f(\theta_1), f(\theta_2) \in \Theta'$ are $(\epsilon, \delta)$-indistinguishable.*

*Proof.* Consider an arbitrary set $T' \subseteq \Theta'$; let $T = \{r \in \Theta : f(r) \in T'\}$. Then, it holds that

$$\begin{aligned} \Pr[f(\theta_1) \in T'] &= \Pr[\theta_1 \in T] \\ &\leq e^\epsilon \Pr[\theta_2 \in T] + \delta \\ &= e^\epsilon \Pr[f(\theta_2) \in T'] + \delta \end{aligned}$$

as desired. □

We now give a certifiable unlearning guarantee for the most naive retraining algorithm for the downstream task, which we mentioned in the main text as Theorem 3.

**Theorem 6** (Unlearning when releasing $\boldsymbol{A}$ and $\boldsymbol{w}$). *For a downstream task $\mathcal{T}$ with loss function $\ell_{\mathcal{T}}$, consider the unlearning algorithm $\mathcal{U}_{head, naive}$ that first runs Algorithm 1 to compute $\tilde{\boldsymbol{A}} = \mathcal{U}_{base}(S_f, \mathcal{A}_{base}(S), T(S))$, where $(\mathcal{A}_{base}, \mathcal{U}_{base})$ perform utility-preserving unlearning (Theorem 2). Then, it fits a head $\boldsymbol{w} = \arg\min_{\boldsymbol{w} \in \mathcal{W}_{head}} \ell_{\mathcal{T}}(\boldsymbol{w}; \tilde{\boldsymbol{A}})$ and returns $\tilde{\boldsymbol{A}}$ and $\boldsymbol{w}$. We assert that $(\mathcal{A}_{head, naive}, \mathcal{U}_{head, naive})$ performs utility-preserving unlearning (Definition 4).*

*Proof.* Intuitively, this is a result of post processing. More precisely, consider the $(\epsilon, \delta)$-indistinguishable base models $\tilde{\boldsymbol{A}} := \mathcal{U}_{base}(S_f, \mathcal{A}_{base}(S), T(S))$ and $\tilde{\boldsymbol{A}}' := \mathcal{U}_{base}(\emptyset, \mathcal{A}_{base}(S \setminus S_f), T(S \setminus S_f))$. Then, since the head fitting is a deterministic post-processing of the original model, this proves the $(\epsilon, \delta)$-indistinguishability between the two.

To prove the utility preservation, observe that in this setting

$$\mathbb{E}[\|\tilde{\boldsymbol{A}} - \boldsymbol{A}^\star\|_\infty] \leq 0.01$$

We thus obtain by Lemma 20

$$\begin{aligned} \mathbb{E}[\|\boldsymbol{w}^\star(\tilde{\boldsymbol{A}}) - \boldsymbol{w}^\star(\boldsymbol{A}^\star)\|_\infty] &\leq \mathbb{E}[\|\boldsymbol{w}^\star(\tilde{\boldsymbol{A}}) - \boldsymbol{w}^\star(\boldsymbol{A}^\star)\|_2] \\ &\leq \frac{L_\infty}{\lambda} \mathbb{E}[\|\tilde{\boldsymbol{A}} - \boldsymbol{A}^\star\|_\infty] \end{aligned}$$

which is at most 0.01, up to constant rescaling. □

The above result is nice, and it follows from the fact that the training algorithm of the downstream task head is just a post-processing. However, a downside is that it still requires retraining of the downstream task head. We can show something stronger: even without provable unlearning of the base model ($\boldsymbol{A}$ and $\boldsymbol{R}$), we can achieve provable unlearning of the downstream task head weights when the downstream task loss is convex in the trainable weights $w$.

We will now consider an arbitrary task $\mathcal{T}$. We first give the following notation.

**Definition 8.** For a base model $\boldsymbol{A}$, let $\boldsymbol{w}^\star(\boldsymbol{A}) := \arg\min_{\boldsymbol{w}} \ell_{\mathcal{T}}(\boldsymbol{w}; \boldsymbol{A})$.

First, we give the following helper lemma that will be useful later on.

**Lemma 20.** *Consider two base models $\boldsymbol{A}_1$ and $\boldsymbol{A}_2$. Then, it holds that*

$$\|\boldsymbol{w}^\star(\boldsymbol{A}_1) - \boldsymbol{w}^\star(\boldsymbol{A}_2)\|_2 \leq \frac{L_\infty}{\lambda}\|\boldsymbol{A}_1 - \boldsymbol{A}_2\|_\infty$$

*Proof.* Observe that

$$
\begin{aligned}
\lambda\|\boldsymbol{w}^\star(\boldsymbol{A}_1) - \boldsymbol{w}^\star(\boldsymbol{A}_2)\|_2 &\leq \|\nabla_{\boldsymbol{w}}\ell_{\mathcal{T}}(\boldsymbol{w}^\star(\boldsymbol{A}_1); \boldsymbol{A}_2) - \nabla_{\boldsymbol{w}}\ell_{\mathcal{T}}(\boldsymbol{w}^\star(\boldsymbol{A}_2); \boldsymbol{A}_2)\|_2 \\
&= \|\nabla_{\boldsymbol{w}}\ell_{\mathcal{T}}(\boldsymbol{w}^\star(\boldsymbol{A}_1); \boldsymbol{A}_2) - \nabla_{\boldsymbol{w}}\ell_{\mathcal{T}}(\boldsymbol{w}^\star(\boldsymbol{A}_1); \boldsymbol{A}_1)\|_2 \\
&\leq L_\infty\|\boldsymbol{A}_1 - \boldsymbol{A}_2\|_\infty
\end{aligned}
$$

where the first line follows from strong convexity, the second line from the gradients being zero, and the third line from the definition of $L_\infty$ Lipschitz constant. Dividing both sides by $\lambda$ gives the desired result. $\square$

We now define the following notations for clarity.

- $\boldsymbol{w}^S := \boldsymbol{w}^\star(\boldsymbol{A}^S)$
- $\boldsymbol{w}^F := \boldsymbol{w}^\star(\boldsymbol{A}^F)$
- $\bar{\boldsymbol{w}}^\star := \boldsymbol{w}^\star(\bar{\boldsymbol{A}})$
- $\bar{\boldsymbol{w}} := \boldsymbol{w}^S - H_{\boldsymbol{w}^S}^{-1}\nabla_w\ell_{\mathcal{T}}(\boldsymbol{w}^S; \bar{A})$, which is the Newton step we take from $\boldsymbol{w}^S$ to approximate $\bar{\boldsymbol{w}}^\star$

First, we give a bound on the approximation error of the Newton step.

**Lemma 21.** *It holds that*

$$\|\bar{\boldsymbol{w}} - \bar{\boldsymbol{w}}^\star\| \leq \frac{L_2 L_\infty^2}{2\lambda^3}\|\boldsymbol{A}^S - \bar{\boldsymbol{A}}\|_\infty^2$$

*Proof.* We aim to bound the distance of the Newton step from $\bar{\boldsymbol{w}}^\star$:

$$\bar{\boldsymbol{w}} - \bar{\boldsymbol{w}}^\star = \left(\boldsymbol{w}^S - H_{\boldsymbol{w}^S}^{-1}\nabla_w\ell_{\mathcal{T}}(\bar{\boldsymbol{A}}, \boldsymbol{w}^S)\right) - \bar{\boldsymbol{w}}^\star$$

where $\boldsymbol{H}_{\boldsymbol{w}^S} = \nabla_{\boldsymbol{w}}^2\ell_{\mathcal{T}}(\bar{\boldsymbol{A}}, \boldsymbol{w}^S)$. Then, it holds that

$$
\begin{aligned}
&\boldsymbol{w}^S - H_{\boldsymbol{w}^S}^{-1}\nabla_w\ell_{\mathcal{T}}(\bar{\boldsymbol{A}}, \boldsymbol{w}^S) - \bar{\boldsymbol{w}}^\star \\
&= \boldsymbol{w}^S - \bar{\boldsymbol{w}}^\star - H_{\boldsymbol{w}^S}^{-1}\left(\nabla_w\ell_{\mathcal{T}}(\bar{\boldsymbol{A}}, \boldsymbol{w}^S) - \nabla_w\ell_{\mathcal{T}}(\bar{\boldsymbol{A}}, \bar{\boldsymbol{w}}^\star)\right) \\
&= H_{\boldsymbol{w}^S}^{-1}\left(H_{\boldsymbol{w}^S}(\boldsymbol{w}^S - \bar{\boldsymbol{w}}^\star) - \int_0^1 H_{\bar{\boldsymbol{w}}^\star + t(\boldsymbol{w}^S - \bar{\boldsymbol{w}}^\star)}(\boldsymbol{w}^S - \bar{\boldsymbol{w}}^\star)dt\right) \\
&= H_{\boldsymbol{w}^S}^{-1}\int_0^1 \left(H_{\boldsymbol{w}^S} - H_{\bar{\boldsymbol{w}}^\star + t(\boldsymbol{w}^S - \bar{\boldsymbol{w}}^\star)}\right)dt \cdot (\boldsymbol{w}^S - \bar{\boldsymbol{w}}^\star)
\end{aligned}
$$

The norm of this quantity is therefore bounded by

$$
\begin{aligned}
&\|H_{\boldsymbol{w}^S}^{-1}\|_2 \cdot \frac{L_2}{2}\|\boldsymbol{w}^S - \bar{\boldsymbol{w}}^\star\| \cdot \|\boldsymbol{w}^S - \bar{\boldsymbol{w}}^\star\| \\
&\leq \frac{L_2}{2\lambda}\|\boldsymbol{w}^S - \bar{\boldsymbol{w}}^\star\|_2^2 \\
&\leq \frac{L_2}{2\lambda}\left(\frac{1}{\lambda}\|\nabla\ell_{\mathcal{T}}(\bar{\boldsymbol{A}}, \boldsymbol{w}^S) - \nabla\ell_{\mathcal{T}}(\boldsymbol{A}^S, \boldsymbol{w}^S)\|_2\right)^2 \\
&\leq \frac{L_2}{2\lambda}\left(\frac{L_\infty}{\lambda}\|\bar{\boldsymbol{A}} - \boldsymbol{A}^S\|_\infty\right)^2
\end{aligned}
$$

Hence, we have that

$$\|\bar{\boldsymbol{w}} - \bar{\boldsymbol{w}}^\star\|_2 \leq \frac{L_2 L_\infty^2}{2\lambda^3}\|\boldsymbol{A}^S - \bar{\boldsymbol{A}}\|_\infty^2$$

$\square$

## C.1 INSTANTIATING FOR $\mathbb{T}_{\text{CLF}} = [r]$

We first instantiate Theorem 4 for the case where $\mathbb{T}_{\text{clf}} = [r]$, or equivalently when $q = 1/ar$.

**Lemma 22.** *Recall our retrained model for the downstream task is $A^F w^F$. Then, it holds that*

$$\|\bar{A}\bar{w} - A^F w^F\|_2 \leq O\left(\sqrt{r}\left(\frac{(ar)^2 m_U}{m\epsilon_0\gamma p}\right)^2 + B\sqrt{nr}\frac{(ar)^2 m_U}{m\epsilon_0\gamma p}\right)$$

*Proof.* We rewrite as follows.

$$\bar{A}\bar{w} - A^F w^F = \left(\bar{A}\bar{w} - \bar{A}\bar{w}^\star\right) + \left(\bar{A}\bar{w}^\star - A^F\bar{w}^\star\right) + \left(A^F\bar{w}^\star - A^F w^F\right)$$

Now, we proceed to bound the $\ell_2$ norm of each of these individual terms separately. For the first term, we have that

$$\begin{aligned}
\|\bar{A}\bar{w} - \bar{A}\bar{w}^\star\|_2 &= \|\bar{A}(\bar{w} - \bar{w}^\star)\|_2 \\
&\leq \|\bar{w} - \bar{w}^\star\|_1 \\
&\leq \sqrt{r}\|\bar{w} - \bar{w}^\star\|_2 \\
&\leq \sqrt{r}\frac{L_2 L_\infty^2}{2\lambda^3}\|A^S - \bar{A}\|_\infty^2 \\
&\leq \sqrt{r}\frac{L_2 L_\infty^2}{2\lambda^3}\left(\frac{(ar)^2 m_U}{m\epsilon_0\gamma p}\right)^2
\end{aligned}$$

where second line follows from $\bar{A}$ having column sum 1, and the fourth line follows from Lemma 20 For the third term, we have a similar analysis.

$$\begin{aligned}
\|A^F\bar{w}^\star - A^F w^F\|_2 &= \|A^F(\bar{w}^\star - w^F)\|_2 \\
&\leq \|\bar{w}^\star - w^F\|_1 \\
&\leq \sqrt{r}\|\bar{w}^\star - w^F\|_2 \\
&\leq \sqrt{r}\frac{L_\infty}{\lambda}\|\bar{A} - A^F\|_\infty \\
&\leq \sqrt{r}\frac{L_\infty}{\lambda}\left(\frac{(ar)^2 m_U}{m\epsilon_0\gamma p}\right)
\end{aligned}$$

Finally, for the second term, we have that

$$\begin{aligned}
\|\bar{A}\bar{w}^\star - A^F\bar{w}^\star\|_2 &\leq \|\bar{A} - A^F\|_2\|\bar{w}^\star\|_2 \\
&\leq \|\bar{A} - A^F\|_\infty \sqrt{nr}\|\bar{w}^\star\|_2 \\
&\leq O\left(\frac{(ar)^2 m_U}{m\epsilon_0\gamma p}\sqrt{nr}B\right)
\end{aligned}$$

By triangle inequality, we obtain the desired result. $\square$

First, we note show the following property of the learned topic model $A^S$.

**Lemma 23.** *The minimum singular value of the ground truth topic matrix $A^S$ is at least $\Theta(p)$, since the perturbations in entries of $A^\star$ are at most $\epsilon_0 \leq O(1/\sqrt{nr})$. Hence, the singular values cannot change by more than a constant factor relative to $p$.*

*Proof.* We know that $A^\star$ is a $p$-separable topic model, and hence has smallest singular value at least $p$. For the given sample complexity of learning, $A^S$ will have smallest singular value at least $\Theta(p)$. $\square$

The above result says that $A^S$ has a unique pseudoinverse, and has largest singular value at most $O(1/p)$.

Recall that our goal for the downstream task is to approximate the $\boldsymbol{v}$ such that

$$\boldsymbol{A}^S \boldsymbol{v} = \boldsymbol{A}^F \boldsymbol{w}^F$$

in order to say we have approximated the unlearned fine-tuned model. Therefore, it suffices to obtain indistinguishability of our unlearning algorithm output $\tilde{\boldsymbol{w}}$ with $(\boldsymbol{A}^S)^\dagger \boldsymbol{A}^F \boldsymbol{w}^F$. Our following claim is that we can use $(\boldsymbol{A}^S)^\dagger \bar{\boldsymbol{A}} \bar{\boldsymbol{w}}$ as the approximation for this.

**Proposition 4.** *It holds that*

$$\|(\boldsymbol{A}^S)^\dagger \bar{\boldsymbol{A}} \bar{\boldsymbol{w}} - (\boldsymbol{A}^S)^\dagger \boldsymbol{A}^F \boldsymbol{w}^F\|_2 \leq O\left(\frac{1}{p} \|\bar{\boldsymbol{A}} \bar{\boldsymbol{w}} - \boldsymbol{A}^F \boldsymbol{w}^F\|_2\right)$$

$$\leq O\left(\frac{1}{p} \cdot \left[\sqrt{r}\left(\frac{(ar)^2 m_U}{m \epsilon_0 \gamma p}\right)^2 + B\sqrt{nr}\frac{(ar)^2 m_U}{m \epsilon_0 \gamma p}\right]\right)$$

Let $\bar{\boldsymbol{v}} := (\boldsymbol{A}^S)^\dagger \bar{\boldsymbol{A}} \bar{\boldsymbol{w}}$ and $\boldsymbol{v} = (\boldsymbol{A}^S)^\dagger \boldsymbol{A}^F \boldsymbol{w}^F$. We claim the following.

**Lemma 24.** *The unlearning algorithm $\mathcal{U}_{head}$ that outputs*

$$\tilde{\boldsymbol{v}} := \bar{\boldsymbol{v}} + \nu_v$$

*where $\nu_v$ is the noise defined by the Gaussian mechanism using the above sensitivity satisfies provable $(\epsilon, \delta)$ unlearning. In particular, we use*

$$\sigma = \frac{O\left(\frac{1}{p} \cdot \left[\sqrt{r}\left(\frac{(ar)^2 m_U}{m \epsilon_0 \gamma p}\right)^2 + B\sqrt{nr}\frac{(ar)^2 m_U}{m \epsilon_0 \gamma p}\right]\right)}{\epsilon} \sqrt{2 \log(1.25/\delta)}$$

*where the numerator of the fraction is from the previous proposition.*

*Proof.* This follows from Gaussian mechanism. $\qquad \square$

We now proceed to bound the deletion capacity. In this case, the utility is defined by the closeness of $\tilde{\boldsymbol{v}}$ to $(\boldsymbol{A}^S)^\dagger \boldsymbol{A}^\star \boldsymbol{w}^\star$ in $\ell_\infty$ norm, similar the way we defined this for the base model unlearning algorithm $\mathcal{U}_{base}$ earlier.

First, the following lemma to bound $\boldsymbol{A}^F \boldsymbol{w}^F - \boldsymbol{A}^\star \boldsymbol{w}^\star$.

**Lemma 25.** *We have that*

$$\|\boldsymbol{A}^F \boldsymbol{w}^F - \boldsymbol{A}^\star \boldsymbol{w}^\star\|_2 \leq O\left(B\sqrt{nr}\frac{(ar)^2 m_U}{m \epsilon_0 \gamma p}\right)$$

*Proof.* We decompose as follows.

$$\boldsymbol{A}^F \boldsymbol{w}^F - \boldsymbol{A}^\star \boldsymbol{w}^\star = (\boldsymbol{A}^F \boldsymbol{w}^F - \boldsymbol{A}^F \boldsymbol{w}^\star) + (\boldsymbol{A}^F \boldsymbol{w}^\star - \boldsymbol{A}^\star \boldsymbol{w}^\star)$$

The first term is bounded by

$$\|\boldsymbol{A}^F \boldsymbol{w}^F - \boldsymbol{A}^F \boldsymbol{w}^\star\|_2 \leq \sqrt{r}\|\boldsymbol{w}^F - \boldsymbol{w}^\star\|_2 \leq O(\sqrt{r}\|\boldsymbol{A}^F - \boldsymbol{A}^\star\|_\infty) \leq O\left(\sqrt{r}\frac{(ar)^2 m_U}{m \epsilon_0 \gamma p}\right)$$

The second term is bounded by

$$\|\boldsymbol{A}^F \boldsymbol{w}^\star - \boldsymbol{A}^\star \boldsymbol{w}^\star\|_2 \leq O\left(\frac{(ar)^2 m_U}{m \epsilon_0 \gamma p}\sqrt{nr}B\right)$$

by considering the spectral norm $\|\boldsymbol{A}^F - \boldsymbol{A}^\star\|_2$. This gives the desired result. $\qquad \square$

As a result, the following holds.

**Proposition 5.** *It holds that*

$$\|(\boldsymbol{A}^S)^\dagger \boldsymbol{A}^F \boldsymbol{w}^F - (\boldsymbol{A}^S)^\dagger \boldsymbol{A}^\star \boldsymbol{w}^\star\|_2 \leq O\left(\frac{1}{p}\left[\sqrt{r}\frac{(ar)^2 m_U}{m \epsilon_0 \gamma p} + B\sqrt{nr}\frac{(ar)^2 m_U}{m \epsilon_0 \gamma p}\right]\right)$$

This is once again from the bounded operator norm property of $(\boldsymbol{A}^S)^\dagger$.

Finally, we can apply triangle inequality to get the following.

**Lemma 26.** *It holds that*

$$\|(\boldsymbol{A}^S)^\dagger \bar{\boldsymbol{A}}\bar{\boldsymbol{w}} - (\boldsymbol{A}^S)^\dagger \boldsymbol{A}^\star \boldsymbol{w}^\star\|_2 \le \left(\frac{1}{p} \cdot \left[\sqrt{r}\left(\frac{(ar)^2 m_U}{m\epsilon_0 \gamma p}\right)^2 + B\sqrt{nr}\frac{(ar)^2 m_U}{m\epsilon_0 \gamma p}\right]\right)$$

Then, we can get the following bound on deletion capacity.

**Lemma 27.** *For $\epsilon, \delta > 0$, the deletion capacity satisfies*

$$T_{\epsilon,\delta}^{\mathcal{A}_{head}, \mathcal{U}_{head}}(m) \ge \tilde{\Omega}\left(\frac{m}{r^2 \sqrt{nr}}\right)$$

*Proof.* The calculation is as follows.

$$\mathbb{E}\big[\|\tilde{\boldsymbol{v}} - (\boldsymbol{A}^S)^\dagger \boldsymbol{A}^\star \boldsymbol{w}^\star\|_\infty\big] \le \mathbb{E}[\|\nu_{\boldsymbol{v}}\|_\infty] + \mathbb{E}\big[\|(\boldsymbol{A}^S)^\dagger \bar{\boldsymbol{A}}\bar{\boldsymbol{w}} - (\boldsymbol{A}^S)^\dagger \boldsymbol{A}^\star \boldsymbol{w}^\star\|_\infty\big]$$

$$\le \left(\frac{1}{p} \cdot \left[\sqrt{r}\left(\frac{(ar)^2 m_U}{m\epsilon_0 \gamma p}\right)^2 + B\sqrt{nr}\frac{(ar)^2 m_U}{m\epsilon_0 \gamma p}\right]\right)\left(\frac{\sqrt{\log r \log 1/\delta}}{\epsilon} + 1\right)$$

For this to be a small constant, we require

$$\frac{(ar)^2 m_U}{m\epsilon_0 \gamma p} \le \tilde{O}\left(\min\left\{\frac{1}{r^{1/4}}, \frac{1}{\sqrt{nr}}\right\}\right)$$

Therefore, we should have

$$m_U \le \tilde{\Omega}\left(\frac{m}{r^2 \sqrt{nr}}\right)$$

$\square$

## C.2 PROOF FOR GENERAL $q$

The following is the formal statement of Theorem 4.

**Theorem 7** (Formal version of Theorem 4). *Suppose that the downstream task $\mathcal{T}$ only depends on a subset of topics $\mathbb{T}_{clf} \subseteq [r]$; that is, $\boldsymbol{w}^\star = \arg\min_{\boldsymbol{v} \in \mathcal{W}_{base}} \ell_{\mathcal{T}}(\boldsymbol{v}; \boldsymbol{A}^\star)$ has non-zero entries only in the index set $\mathbb{T}_{clf}$. Denote $q := \min_{k \in \mathbb{T}_{clf}} \Pr_{\mathcal{D}}[z = k]$, and let $\mathcal{A}_{head}$ be the head tuning algorithm (Definition 2) and $\mathcal{U}_{head}$ be Algorithm 2. Then, $(\mathcal{A}_{head}, \mathcal{U}_{head})$ performs utility-preserving unlearning with deletion capacity*

$$T_{\epsilon,\delta}^{\mathcal{A}_{head}, \mathcal{U}_{head}}(m) \ge c' \cdot \min\left\{\frac{mq\epsilon}{r\sqrt{nr \log 1/\delta}}, \frac{0.001m}{r^2}\right\}$$

*where $c'$ is a constant dependent on $\mathcal{D}$, and $\mathcal{T}$.*

**Lemma 28.** *Recall our retrained model for the downstream task is $\boldsymbol{A}^F w^F$. Then, it holds that*

$$\|\bar{\boldsymbol{A}}\bar{\boldsymbol{w}} - \boldsymbol{A}^F \boldsymbol{w}^F\|_2 \le O\left(\sqrt{r}\left(\frac{(ar)^2 m_U}{m\epsilon_0 \gamma p}\right)\right) + O\left(B\sqrt{nr}\frac{(1/q)arm_U}{m\epsilon_0 \gamma p}\right) + O\left(\left(\frac{(ar)^2 m_U}{m\epsilon_0 \gamma p}\right)^2 \sqrt{nr}\right)$$

*Proof.* Consider this decomposition again.

$$\bar{\boldsymbol{A}}\bar{\boldsymbol{w}} - \boldsymbol{A}^F \boldsymbol{w}^F = \left(\bar{\boldsymbol{A}}\bar{\boldsymbol{w}} - \bar{\boldsymbol{A}}\bar{\boldsymbol{w}}^\star\right) + \left(\bar{\boldsymbol{A}}\bar{\boldsymbol{w}}^\star - \boldsymbol{A}^F \bar{\boldsymbol{w}}^\star\right) + \left(\boldsymbol{A}^F \bar{\boldsymbol{w}}^\star - \boldsymbol{A}^F \boldsymbol{w}^F\right)$$

The first term is the same as old analysis; the second term is from considering $q$; the third is the same as the old analysis. In particular, when $q = 1/ar$, we recover the old bound. We have that the first term is

$$\|\bar{\boldsymbol{A}}\bar{\boldsymbol{w}} - \bar{\boldsymbol{A}}\bar{\boldsymbol{w}}^\star\| \le \sqrt{r}\frac{L_2 L_\infty^2}{2\lambda^3}\left(\frac{(ar)^2 m_U}{m\epsilon_0 \gamma p}\right)^2$$

The third term is

$$\|\boldsymbol{A}^F \bar{\boldsymbol{w}}^\star - \boldsymbol{A}^F \boldsymbol{w}^F\| \le \sqrt{r} \frac{L_\infty}{\lambda} \left( \frac{(ar)^2 m_U}{m\epsilon_0 \gamma p} \right)$$

The second term is

$$\|\bar{\boldsymbol{A}} \bar{\boldsymbol{w}}^\star - \boldsymbol{A}^F \bar{\boldsymbol{w}}^\star\| \le \|(\bar{\boldsymbol{A}} - \boldsymbol{A}^F) \bar{\boldsymbol{w}}^\star\| + \|(\bar{\boldsymbol{A}} - \boldsymbol{A}^F)(\boldsymbol{w}^\star - \bar{\boldsymbol{w}}^\star)\|$$

$$\le O\left( B\sqrt{nr} \frac{(1/q)arm_U}{m\epsilon_0 \gamma p} \right) + O\left( \left( \frac{(ar)^2 m_U}{m\epsilon_0 \gamma p} \right)^2 \sqrt{nr} \right)$$

This gives the desired result using triangle inequality. $\qquad\square$

Continuing, we have the following.

**Proposition 6.** *It holds that*

$$\|(\boldsymbol{A}^S)^\dagger \bar{\boldsymbol{A}} \bar{\boldsymbol{w}} - (\boldsymbol{A}^S)^\dagger \boldsymbol{A}^F \boldsymbol{w}^F\|_2$$

$$\le O\left( \frac{1}{p} \|\bar{\boldsymbol{A}} \bar{\boldsymbol{w}} - \boldsymbol{A}^F \boldsymbol{w}^F\|_2 \right)$$

$$\le O\left( \frac{1}{p} \cdot \left[ \sqrt{r} \left( \frac{(ar)^2 m_U}{m\epsilon_0 \gamma p} \right) + B\sqrt{nr} \frac{(1/q)arm_U}{m\epsilon_0 \gamma p} + \left( \frac{(ar)^2 m_U}{m\epsilon_0 \gamma p} \right)^2 \sqrt{nr} \right] \right)$$

This gives us the following.

**Lemma 29.** *The unlearning algorithm $\mathcal{U}_{head}$ that outputs*

$$\tilde{\boldsymbol{v}} := \bar{\boldsymbol{v}} + \nu_v$$

*where $\nu_v$ is the noise defined by the Gaussian mechanism using the above sensitivity satisfies provable $(\epsilon, \delta)$ unlearning. In particular, we use*

$$\sigma = \frac{O\left( \frac{1}{p} \cdot \left[ \sqrt{r} \left( \frac{(ar)^2 m_U}{m\epsilon_0 \gamma p} \right) + B\sqrt{nr} \frac{(1/q)arm_U}{m\epsilon_0 \gamma p} + \left( \frac{(ar)^2 m_U}{m\epsilon_0 \gamma p} \right)^2 \sqrt{nr} \right] \right)}{\epsilon} \sqrt{2\log(1.25/\delta)}$$

*where the numerator of the fraction is from the previous proposition.*

*Proof.* This follows from Gaussian mechanism. $\qquad\square$

We now proceed to bound the deletion capacity. In this case, the utility is defined by the closeness of $\tilde{\boldsymbol{v}}$ to $(\boldsymbol{A}^S)^\dagger \boldsymbol{A}^\star \boldsymbol{w}^\star$ in $\ell_\infty$ norm, similar the way we defined this for the base model unlearning algorithm $\mathcal{U}_{base}$ earlier.

First, the following lemma to bound $\boldsymbol{A}^F \boldsymbol{w}^F - \boldsymbol{A}^\star \boldsymbol{w}^\star$.

**Lemma 30.** *We have that*

$$\|\boldsymbol{A}^F \boldsymbol{w}^F - \boldsymbol{A}^\star \boldsymbol{w}^\star\|_2 \le O\left( \sqrt{r} \left( \frac{(ar)^2 m_U}{m\epsilon_0 \gamma p} \right) + B\sqrt{nr} \frac{(1/q)arm_U}{m\epsilon_0 \gamma p} + \left( \frac{(ar)^2 m_U}{m\epsilon_0 \gamma p} \right)^2 \sqrt{nr} \right)$$

*Proof.* We decompose as follows.

$$\boldsymbol{A}^F \boldsymbol{w}^F - \boldsymbol{A}^\star \boldsymbol{w}^\star = (\boldsymbol{A}^F \boldsymbol{w}^F - \boldsymbol{A}^F \boldsymbol{w}^\star) + (\boldsymbol{A}^F \boldsymbol{w}^\star - \boldsymbol{A}^\star \boldsymbol{w}^\star)$$

The first term is bounded by

$$\|\boldsymbol{A}^F \boldsymbol{w}^F - \boldsymbol{A}^F \boldsymbol{w}^\star\|_2 \le \sqrt{r} \|\boldsymbol{w}^F - \boldsymbol{w}^\star\|_2 \le O(\sqrt{r} \|\boldsymbol{A}^F - \boldsymbol{A}^\star\|_\infty) \le O\left( \sqrt{r} \left( \frac{(ar)^2 m_U}{m\epsilon_0 \gamma p} \right) \right)$$

The second term is bounded by

$$\|\boldsymbol{A}^F \boldsymbol{w}^\star - \boldsymbol{A}^\star \boldsymbol{w}^\star\|_2 \le B\sqrt{nr} \frac{(1/q)arm_U}{m\epsilon_0 \gamma p} + \left( \frac{(ar)^2 m_U}{m\epsilon_0 \gamma p} \right)^2 \sqrt{nr}$$

Triangle inequality gives us the desired result. $\qquad\square$

As a result, the following holds.

**Proposition 7.** *It holds that*

$$\|(\boldsymbol{A}^S)^\dagger \boldsymbol{A}^F \boldsymbol{w}^F - (\boldsymbol{A}^S)^\dagger \boldsymbol{A}^\star \boldsymbol{w}^\star\|_2 \leq O\left(\frac{1}{p} \cdot \left[\sqrt{r}\left(\frac{(ar)^2 m_U}{m\epsilon_0\gamma p}\right) + B\sqrt{nr}\frac{(1/q)arm_U}{m\epsilon_0\gamma p} + \left(\frac{(ar)^2 m_U}{m\epsilon_0\gamma p}\right)^2 \sqrt{nr}\right]\right)$$

This is once again from the bounded operator norm property.

Finally, we can apply triangle inequality to get the following.

**Lemma 31.** *It holds that*

$$\|(\boldsymbol{A}^S)^\dagger \bar{\boldsymbol{A}}\bar{\boldsymbol{w}} - (\boldsymbol{A}^S)^\dagger \boldsymbol{A}^\star \boldsymbol{w}^\star\|_2 \leq O\left(\frac{1}{p} \cdot \left[\sqrt{r}\left(\frac{(ar)^2 m_U}{m\epsilon_0\gamma p}\right) + B\sqrt{nr}\frac{(1/q)arm_U}{m\epsilon_0\gamma p} + \left(\frac{(ar)^2 m_U}{m\epsilon_0\gamma p}\right)^2 \sqrt{nr}\right]\right)$$

Then, we can get the following bound on deletion capacity.

**Lemma 32.** *For $\epsilon, \delta > 0$, the deletion capacity satisfies*

$$T_{\epsilon,\delta}^{\mathcal{A}_{head}, \mathcal{U}_{head}}(m) \geq \tilde{\Omega}\left(\frac{m}{r^2\sqrt{nr}}\right)$$

*Proof.* The calculation is as follows.

$$\mathbb{E}\big[\|\tilde{\boldsymbol{v}} - (\boldsymbol{A}^S)^\dagger \boldsymbol{A}^\star \boldsymbol{w}^\star\|_\infty\big] \leq \mathbb{E}[\|\nu_{\boldsymbol{v}}\|_\infty] + \mathbb{E}\big[\|(\boldsymbol{A}^S)^\dagger \bar{\boldsymbol{A}}\bar{\boldsymbol{w}} - (\boldsymbol{A}^S)^\dagger \boldsymbol{A}^\star \boldsymbol{w}^\star\|_\infty\big]$$

$$\leq \left(\frac{1}{p} \cdot \left[\sqrt{r}\left(\frac{(ar)^2 m_U}{m\epsilon_0\gamma p}\right) + B\sqrt{nr}\frac{(1/q)arm_U}{m\epsilon_0\gamma p} + \left(\frac{(ar)^2 m_U}{m\epsilon_0\gamma p}\right)^2 \sqrt{nr}\right]\right)$$

$$\cdot \left(\frac{\sqrt{\log r \log 1/\delta}}{\epsilon} + 1\right)$$

For this to be a small constant, we require

$$\frac{(ar)^2 m_U}{m\epsilon_0\gamma p} \leq \tilde{O}\left(\min\left\{\frac{1}{r^{1/2}}, \frac{1}{(nr)^{1/4}}, \frac{arq}{\sqrt{nr}}\right\}\right)$$

When $n$ is at least $r^3$, the last of these terms will be the smallest. Therefore, we have that

$$m_U \leq \tilde{\Omega}\left(\frac{mq}{r^{1.5}n^{0.5}}\right)$$

$\square$

## C.3 DOWNSTREAM TASKS WITH INEXACT MINIMIZERS

We will consider $\tau$-optimal minimizers for the downstream task.

**Definition 9.** For a base model $\boldsymbol{A}$, let $\boldsymbol{w}_\tau(\boldsymbol{A}) \in \{\boldsymbol{w} : \ell_\mathcal{T}(\boldsymbol{w}; \boldsymbol{A}) - \ell_\mathcal{T}(\boldsymbol{w}^\star(\boldsymbol{A}); \boldsymbol{A}) \leq \tau\}$.

We will have the following technical assumption on $\tau$ to eliminate trivial cases.

**Assumption 8.** *We assume that $\tau = O(1/r)$.*

We have the following intermediate result.

**Lemma 33.** *For a base model $\boldsymbol{A}$, it holds that*

$$\|\boldsymbol{w}_\tau(\boldsymbol{A}) - \boldsymbol{w}^\star(\boldsymbol{A})\|_2 \leq \sqrt{2\tau/\lambda}$$

*Proof.* This follows from a standard strong convexity argument. $\square$

This gives an updated version of Lemma 20, using triangle inequality.

**Lemma 34.** *Consider two base models $\boldsymbol{A}_1$ and $\boldsymbol{A}_2$. Then, it holds that*

$$\|\boldsymbol{w}_\tau(\boldsymbol{A}_1) - \boldsymbol{w}_\tau(\boldsymbol{A}_2)\|_2 \le 2\sqrt{2\tau/\lambda} + \|\boldsymbol{w}^\star(\boldsymbol{A}_1) - \boldsymbol{w}^\star(\boldsymbol{A}_2)\| \le 2\sqrt{2\tau/\lambda} + \frac{L_\infty}{\lambda}\|\boldsymbol{A}_1 - \boldsymbol{A}_2\|_\infty$$

We now define the following notations for clarity.

- $\boldsymbol{w}_\tau^S := \boldsymbol{w}_\tau(\boldsymbol{A}^S)$
- $\boldsymbol{w}_\tau^F := \boldsymbol{w}_\tau(\boldsymbol{A}^F)$
- $\bar{\boldsymbol{w}}^\star := \boldsymbol{w}^\star(\bar{\boldsymbol{A}})$
- $\bar{\boldsymbol{w}}_\tau := \boldsymbol{w}_\tau^S - H_{\boldsymbol{w}_\tau^S}^{-1}\nabla_w \ell_{\mathcal{T}}(\boldsymbol{w}_\tau^S; \bar{A})$, which is the Newton step we take from $\boldsymbol{w}_\tau^S$ to approximate $\bar{\boldsymbol{w}}^\star$

Now, consider our Newton step from $\boldsymbol{w}_\tau(\boldsymbol{A}^S)$ to approximate $\boldsymbol{w}_\tau(\boldsymbol{A}^F)$. To do so, consider

$$\bar{\boldsymbol{w}}_\tau := \boldsymbol{w}_\tau^S - H_{\boldsymbol{w}_\tau^S}^{-1}\nabla_w \ell_{\mathcal{T}}(\boldsymbol{w}_\tau^S; \bar{A})$$

Then, the following holds, using a similar argument as a previous section.

$$\begin{aligned}
\|\bar{\boldsymbol{w}}_\tau - \bar{\boldsymbol{w}}^\star\|_2 &\le \frac{L_2}{2\lambda}\|\boldsymbol{w}_\tau^S - \bar{\boldsymbol{w}}^\star\|_2^2 \\
&\le \frac{L_2}{\lambda}\left(\|\boldsymbol{w}_\tau^S - \boldsymbol{w}^S\|_2^2 + \|\boldsymbol{w}^S - \bar{\boldsymbol{w}}^\star\|_2^2\right) \\
&\le \frac{L_2}{\lambda}\left(\frac{2\tau}{\lambda} + \left(\frac{L_\infty}{\lambda}\|\bar{\boldsymbol{A}} - \boldsymbol{A}^S\|_\infty\right)^2\right)
\end{aligned}$$

where the second term of the last line follows from a line in the previous proof of the Newton step.

We now bound $\|\bar{\boldsymbol{A}}\bar{\boldsymbol{w}}_\tau - \boldsymbol{A}^F\boldsymbol{w}_\tau^F\|_2$. We rewrite this as follows.

$$\bar{\boldsymbol{A}}\bar{\boldsymbol{w}}_\tau - \boldsymbol{A}^F\bar{\boldsymbol{w}}_\tau^F = (\bar{\boldsymbol{A}}\bar{\boldsymbol{w}}_\tau - \bar{\boldsymbol{A}}\bar{\boldsymbol{w}}^\star) + (\bar{\boldsymbol{A}}\bar{\boldsymbol{w}}^\star - \boldsymbol{A}^F\boldsymbol{w}^F) + (\boldsymbol{A}^F\boldsymbol{w}^F - \boldsymbol{A}^F\boldsymbol{w}_\tau^F)$$

The second term we bounded as part of a previous proof. The first term we bound using the above, and the third part we bound using the inexact minimizer lemma. By a step in a previous proof, we can say that the first term

$$\begin{aligned}
\|\bar{\boldsymbol{A}}\bar{\boldsymbol{w}}_\tau - \bar{\boldsymbol{A}}\bar{\boldsymbol{w}}^\star\|_2 &\le \sqrt{r}\|\bar{\boldsymbol{w}}_\tau - \bar{\boldsymbol{w}}^\star\|_2 \\
&\le \sqrt{r}\left(\frac{2L_2\tau}{\lambda^2} + \frac{L_2 L_\infty^2}{2\lambda^3}\left(\frac{(ar)^2 m_U}{m\epsilon_0\gamma p}\right)^2\right)
\end{aligned}$$

The second term, to reiterate, satisfies

$$\|\bar{\boldsymbol{A}}\bar{\boldsymbol{w}}^\star - \boldsymbol{A}^F\boldsymbol{w}^F\|_2 \le O\left(\frac{(ar)^2 m_U}{m\epsilon_0\gamma p}\sqrt{nr}B\right)$$

The third term satisfies

$$\|\boldsymbol{A}^F\boldsymbol{w}^F - \boldsymbol{A}^F\boldsymbol{w}_\tau^F\|_2 \le \sqrt{r}\|\boldsymbol{w}^F - \boldsymbol{w}_\tau^F\|_2 \le \sqrt{2r\tau/\lambda}$$

Therefore, the entire term is bounded by the sum of the above three terms via triangle inequality. In particular, this is

$$O\left(\sqrt{r}\left(\frac{(ar)^2 m_U}{m\epsilon_0\gamma p}\right)^2 + B\sqrt{nr}\frac{(ar)^2 m_U}{m\epsilon_0\gamma p} + \sqrt{r\tau} + \sqrt{r\tau}\right)$$

Recall that our goal for the downstream task is to approximate the $\boldsymbol{v}$ such that

$$\boldsymbol{A}^S\boldsymbol{v} = \boldsymbol{A}^F\boldsymbol{w}_\tau^F$$

or in other words, $\boldsymbol{v} = (\boldsymbol{A}^S)^\dagger \boldsymbol{A}^F\boldsymbol{w}_\tau^F$. Our claim is that we can use $(\boldsymbol{A}^S)^\dagger \bar{\boldsymbol{A}}\bar{\boldsymbol{w}}_\tau$ as an approximation for this. The following statement formalizes this, utilizing the fact about the maximum singular value of $\boldsymbol{A}^S$ being at most $\Theta(1/p)$.

**Proposition 8.** *It holds that*

$$\|(\boldsymbol{A}^S)^\dagger \bar{\boldsymbol{A}}\bar{\boldsymbol{w}}_\tau - (\boldsymbol{A}^S)^\dagger \boldsymbol{A}^F \boldsymbol{w}_\tau^F\|_2 \le O\left(\frac{1}{p}\left[\sqrt{r}\left(\frac{(ar)^2 m_U}{m\epsilon_0\gamma p}\right)^2 + B\sqrt{nr}\frac{(ar)^2 m_U}{m\epsilon_0\gamma p} + \sqrt{r\tau} + \sqrt{r\tau}\right]\right)$$

Now, let $\bar{\boldsymbol{v}} := (\boldsymbol{A}^S)^\dagger \bar{\boldsymbol{A}}\bar{\boldsymbol{w}}_\tau$, and let $\boldsymbol{v} := (\boldsymbol{A}^S)^\dagger \boldsymbol{A}^F \boldsymbol{w}_\tau^F$. We claim the following.

**Lemma 35.** *The unlearning algorithm $\mathcal{U}_{head}$ that outputs*

$$\tilde{\boldsymbol{v}} := \bar{\boldsymbol{v}} + \nu_v$$

*where $\nu_v$ is the noise defined by the Gaussian mechanism using the above sensitivity satisfies provable $(\epsilon, \delta)$ unlearning. In particular, we use*

$$\sigma = \frac{O\left(\frac{1}{p}\left[\sqrt{r}\left(\frac{(ar)^2 m_U}{m\epsilon_0\gamma p}\right)^2 + B\sqrt{nr}\frac{(ar)^2 m_U}{m\epsilon_0\gamma p} + \sqrt{r\tau} + \sqrt{r\tau}\right]\right)}{\epsilon}\sqrt{2\log(1.25/\delta)}$$

*where the numerator of the fraction is from the previous proposition.*

*Proof.* This follows from Gaussian mechanism. $\square$

We now proceed to bound the deletion capacity. In this case, the utility is defined by the closeness of $\tilde{\boldsymbol{v}}$ to $(\boldsymbol{A}^S)^\dagger \boldsymbol{A}^\star \boldsymbol{w}^\star$ in $\ell_\infty$ norm, similar the way we defined this for the base model unlearning algorithm $\mathcal{U}_{base}$ earlier.

First, the following lemma to bound $\boldsymbol{A}^F \boldsymbol{w}_\tau^F - \boldsymbol{A}^\star \boldsymbol{w}^\star$.

**Lemma 36.** *We have that*

$$\|\boldsymbol{A}^F \boldsymbol{w}_\tau^F - \boldsymbol{A}^\star \boldsymbol{w}^\star\|_2 \le O\left(B\sqrt{nr}\frac{(ar)^2 m_U}{m\epsilon_0\gamma p} + \sqrt{r\tau}\right)$$

*Proof.* We decompose as follows.

$$\boldsymbol{A}^F \boldsymbol{w}_\tau^F - \boldsymbol{A}^\star \boldsymbol{w}^\star = (\boldsymbol{A}^F \boldsymbol{w}_\tau^F - \boldsymbol{A}^F \boldsymbol{w}^F) + (\boldsymbol{A}^F \boldsymbol{w}^F - \boldsymbol{A}^F \boldsymbol{w}^\star) + (\boldsymbol{A}^F \boldsymbol{w}^\star - \boldsymbol{A}^\star \boldsymbol{w}^\star)$$

The second term is bounded by

$$\|\boldsymbol{A}^F \boldsymbol{w}^F - \boldsymbol{A}^F \boldsymbol{w}^\star\|_2 \le \sqrt{r}\|\boldsymbol{w}^F - \boldsymbol{w}^\star\|_2 \le O(\sqrt{r}\|\boldsymbol{A}^F - \boldsymbol{A}^\star\|_\infty) \le O\left(\sqrt{r}\frac{(ar)^2 m_U}{m\epsilon_0\gamma p}\right)$$

The third term is bounded by

$$\|\boldsymbol{A}^F \boldsymbol{w}^\star - \boldsymbol{A}^\star \boldsymbol{w}^\star\|_2 \le O\left(\frac{(ar)^2 m_U}{m\epsilon_0\gamma p}\sqrt{nr}B\right)$$

by considering the spectral norm $\|\boldsymbol{A}^F - \boldsymbol{A}^\star\|_2$.

Finally, the first term is bounded, from an above argument, by $\sqrt{2r\tau/\lambda}$. This gives the desired result. $\square$

As a result, the following holds.

**Proposition 9.** *It holds that*

$$\|(\boldsymbol{A}^S)^\dagger \boldsymbol{A}^F \boldsymbol{w}_\tau^F - (\boldsymbol{A}^S)^\dagger \boldsymbol{A}^\star \boldsymbol{w}^\star\|_2 \le O\left(\frac{1}{p}\left[\sqrt{r}\frac{(ar)^2 m_U}{m\epsilon_0\gamma p} + B\sqrt{nr}\frac{(ar)^2 m_U}{m\epsilon_0\gamma p} + \sqrt{r\tau}\right]\right)$$

This is once again from the bounded operator norm property of $(\boldsymbol{A}^S)^\dagger$.

Finally, we can apply triangle inequality to get the following.

**Lemma 37.** *It holds that*

$$\|(\boldsymbol{A}^S)^\dagger \bar{\boldsymbol{A}} \bar{\boldsymbol{w}}_\tau - (\boldsymbol{A}^S)^\dagger \boldsymbol{A}^\star \boldsymbol{w}^\star\|_2 \leq \left( \frac{1}{p} \cdot \left[ \sqrt{r} \left( \frac{(ar)^2 m_U}{m\epsilon_0 \gamma p} \right)^2 + B\sqrt{nr} \frac{(ar)^2 m_U}{m\epsilon_0 \gamma p} + \sqrt{r\tau} + \sqrt{r\tau} \right] \right)$$

Then, we can get the following bound on deletion capacity.

**Lemma 38.** *For $\epsilon, \delta > 0$, the deletion capacity satisfies*

$$T_{\epsilon,\delta}^{\mathcal{A}_{head}, \mathcal{U}_{head}}(m) \geq \tilde{\Omega}\left( \frac{m}{r^2 \sqrt{nr}} \right)$$

*Proof.* The calculation is as follows.

$$\mathbb{E}\left[ \|\tilde{\boldsymbol{v}} - (\boldsymbol{A}^S)^\dagger \boldsymbol{A}^\star \boldsymbol{w}^\star\|_\infty \right] \leq \mathbb{E}[\|\nu_{\boldsymbol{v}}\|_\infty] + \mathbb{E}\left[ \|(\boldsymbol{A}^S)^\dagger \bar{\boldsymbol{A}} \bar{\boldsymbol{w}}_\tau - (\boldsymbol{A}^S)^\dagger \boldsymbol{A}^\star \boldsymbol{w}^\star\|_\infty \right]$$

$$\leq \left( \frac{1}{p} \cdot \left[ \sqrt{r} \left( \frac{(ar)^2 m_U}{m\epsilon_0 \gamma p} \right)^2 + B\sqrt{nr} \frac{(ar)^2 m_U}{m\epsilon_0 \gamma p} + \sqrt{r\tau} + \sqrt{r\tau} \right] \right) \left( \frac{\sqrt{\log r \log 1/\delta}}{\epsilon} + 1 \right)$$

For this to be a small constant, we require

$$\frac{(ar)^2 m_U}{m\epsilon_0 \gamma p} \leq \tilde{O}\left( \min\left\{ \frac{1}{r^{1/4}}, \frac{1}{\sqrt{nr}} \right\} \right)$$

since we already have $\tau = O(1/r)$ by assumption. Therefore, we should have

$$m_U \leq \tilde{\Omega}\left( \frac{m}{r^2 \sqrt{nr}} \right)$$

$\square$

Finally, for general $q$, the same idea used in this section for inexact minimizers can be combined with the previous section.

