# OpenReview forum: "Provable unlearning in topic modeling and downstream tasks"
_ICLR.cc/2025/Conference — ICLR 2025 Poster_

### Official Review · Reviewer_JM47 · 2024-11-01

**Soundness:** 3
**Presentation:** 4
**Contribution:** 1
**Rating:** 3
**Confidence:** 4

**Summary:**

This paper combines old with new: an old theoretical bag-of-words topic-model that has provable guarantees, with a new problem: unlearning. It designs unlearning algorithms and proves theorems about these, that in certain cases unlearning can be done using less time than it would take to re-run the learning algorithm.

**Strengths:**

This paper is very well written and addresses a problem that is in the spotlight: unlearning. I haven't checked the math carefully, but it looks like it is likely correct.

**Weaknesses:**

I'm sorry to say it, but bag-of-words language models have little remaining relevance today. Furthermore, in the few cases where they are relevant, they are quite fast to train. Therefore, proving complex theorems about speeding them up is not of general interest. It also does not seem likely that the takeaways from these bag-of-words models provide insight into the larger relevant models. This paper might be of interest to specialized theoretical conferences such as ALT.

**Questions:**

To be honest, this may have been an excellent paper 5-10 years ago, but I'm curious what someone can learn from it today.

---

> ### Author Response · Authors · 2024-11-19
> **Response to Reviewer JM47**
>
> Thank you for your review; below, we respond to the main concerns you may have had regarding our paper. Please let us know if they have been suitably addressed.
>
> We focus on the topic modeling setting because it provides one of the simplest and most tractable frameworks for analyzing pre-training and fine-tuning on text data. Topic modeling is among the few settings with provable learning guarantees for efficient algorithms that perform well on real-world datasets, making it an especially useful model for studying unlearning in the context of the modern day shift towards the pre-training and fine-tuning paradigm.
>
> > **Bag-of-words language models have little remaining relevance today.**
>
> We respectfully disagree with the reviewer. Topic models are used extensively in modern-day interdisciplinary research spanning many fields [1], including medicine [2], social science [3], and the analysis of meta-trends in research [4]. The citations provided are from recent years to emphasize the modern-day relevance, though there are many more classical studies in these areas. Within machine learning, a highly salient application of bag-of-words models is to construct sentence embeddings that can be used for retrieval, recommendation, and search. Indeed, one strong retrieval baseline, which often outperforms large LMs [5], is BM25, which uses a bag-of-words representation of documents [6]. In general, retrieval-augmented systems constitute a highly active and broadly impactful area of research [7], and substantial effort has been invested in trying to ensure privacy in retrieval-augmented settings [8, 9].
>
> > **It also does not seem likely that the takeaways from these bag-of-words models provide insight into the larger relevant models.**
>
> We also reiterate here why we believe the topic model setting provides insight into modern-day LLMs. The $\mathbf{A}$ matrix in the topic model essentially embeds every context (a document, a sentence, or even a partial sentence) into a vector representation. The $\mathbf{X}$ matrix uses that representation to output a distribution over words. This is analogous to language modeling, both masked (e.g., BERT) and autoregressive (e.g., GPT), in that the model backbone (analogous to $\mathbf{A}$) outputs a low-dimensional representation that is multiplied by word embeddings (analogous to $\mathbf{X}$) to produce logits over the word that can follow a given context. We conclude by noting that a number of papers adopt this view of deep learning models to derive meaningful empirical phenomena [10, 11].
>
> References:
>
> [1] Abdelrazek et al., Topic Modeling Algorithms and Applications: A Survey. *Information Systems*, Vol 112 (2023).
>
> [2] Woodman, R.J., Mangoni, A.A. A comprehensive review of machine learning algorithms and their application in geriatric medicine: present and future. *Aging Clin Exp Res* 35, 2363–2397 (2023).
>
> [3] Laureate, C.D.P., Buntine, W. & Linger, H. A systematic review of the use of topic models for short text social media analysis. *Artif Intell Rev* 56, 14223–14255 (2023).
>
> [4] Hwang, S., Flavin, E. & Lee, JE. Exploring research trends of technology use in mathematics education: A scoping review using topic modeling. *Educ Inf Technol* 28, 10753–10780 (2023).
>
> [5] Thakur et al., BEIR: A Heterogeneous Benchmark for Zero-shot Evaluation of Information Retrieval Models (NeurIPS 2021).
>
> [6] Robertson & Zaragoza, The Probabilistic Relevance Framework: BM25 and Beyond (2009).
>
> [7] Gao et al., Retrieval-Augmented Generation for Large Language Models: A Survey (2024).
>
> [8] Min et al., SILO Language Models: Isolating Legal Risk in a Nonparametric Datastore (ICLR 2024).
>
> [9] Huang et al., Privacy Implications of Retrieval-Based Language Models (EMNLP 2023).
>
> [10] Saunshi et al., A Mathematical Exploration of Why Language Models Help Solve Downstream Tasks (ICLR 2021).
>
> [11] Mixon et al., Neural collapse with unconstrained features. Sampling Theory, Signal Processing, and Data Analysis (2022).

---

> > ### Comment · Reviewer_JM47 · 2024-11-22
> >
> > Certainly there are papers and systems that use bag of words models, but their use is rapidly declining (especially within the ICLR community, I believe) and ICLR may want to devote its limited space to other directions.

---

> > > ### Comment · Reviewer_JM47 · 2024-11-27
> > >
> > > I will just add that while I did not find the topic of sufficient interest/fit for the conference, I did not see any technical problems with the paper. Therefore, if the AC and other reviewers all feel that it will be of interest to the ICLR audience, I would be in the minority and it would be worth accepting.

---

> > > > ### Author Response · Authors · 2024-12-02
> > > >
> > > > We thank the reviewer for their efforts in reviewing the paper and appreciate the acknowledgement of the technical soundness of our work. To that end, we kindly ask the reviewer to consider raising their score, if their concerns have been suitably addressed.

---

### Official Review · Reviewer_QxzF · 2024-11-02

**Soundness:** 3
**Presentation:** 3
**Contribution:** 3
**Rating:** 6
**Confidence:** 2

**Summary:**

The paper proposes and analyzes unlearning algorithms in topic language models. In a language model, unlearning algorithms are algorithms to forget a specified set of documents $S_f$ present in training data (used in pre-training or fine-tuning stages). Ideally, once the unlearning is done, the model should be equivalent to one trained without $S_f$.

In the context of the paper, topic modeling means, learning $A: n \times r$ whose columns are word probability distributions for the topics and $R: r \times r $: a topic-topic co-occurence matrix. Here $n$ is the vocabulary  size and $r$ is the number of topics.

The key components of the unlearning algorithms are (a) a Hessian update to the $n$ word-topic distribution vectors $C_i$ each of size $r$ (b) update A, R  and finally (c) introduce DP-noise with Gaussian mechanism.
The key contributions of the paper are to show DP guarantees on the proposed unlearning algorithms.

**Strengths:**

* The paper gives solid theoretical results.

**Weaknesses:**

Unlearning algorithms for language models have practical applications, as the authors argued in the introduction of the paper. However, it is an indeed a limitation — as the authors rightly point out — that the analysis is on topic models, which are not state of the art at this time. Further — I am saying this without citations — Gaussian mechanism is also not practical as it introduces too much noise to make the output useful.

**Questions:**

Footnote in page 4 says that L = 2; that is, the document length is fixed to 2 words. That seems limiting.

---

> ### Author Response · Authors · 2024-11-19
> **Response to Reviewer QxzF**
>
> Thank you for your constructive review! We respond to your concerns below, and please let us know if there is anything else we can clarify further on.
>
> > **Q1: Gaussian mechanism is also not practical as it introduces too much noise to make the output useful.**
>
> **A1**: Generally speaking, the Gaussian mechanism or a variant is necessary for differential privacy based guarantees. Alternative mechanisms, such as those based on correlated noise, can achieve similar guarantees with less noise but typically rely on specific problem structures, which are beyond the scope of this work. Exploring such structures or devising alternative privacy notions that reduce noise while satisfying the notion of unlearning is a valuable direction for future work. We appreciate your concern and agree that balancing privacy and utility is an important challenge deserving further attention.
>
> > **Q2: Footnote in page 4 says that L = 2; that is, the document length is fixed to 2 words. That seems limiting.**
>
> **A2**: Actually, the document length can be generalized to any constant sized amount; the asymptotic deletion capacity will remain the same. As we mentioned in the footnote, the $L = 2$ is simply without loss of generality; if we extend it to some other constant greater than 2, one can obtain the same guarantees via a slight modification to the proof of Lemma 10 in the appendix. We will clarify this further in a future revision.

---

> > ### Comment · Reviewer_QxzF · 2024-11-28
> >
> > I thank the authors for responding to my comments. I will keep my score unchanged.

---

### Official Review · Reviewer_Hp5C · 2024-11-04

**Soundness:** 4
**Presentation:** 3
**Contribution:** 3
**Rating:** 8
**Confidence:** 3

**Summary:**

The paper studies machine unlearning for topic models. Topic modeling is proposed as one of the simplest case of the pretrain/finetune paradigm, where the distribution of words for each topic can be "pretrained" using provable algorithms (Arora et al., 2012), and this matrix can then be adapted for downstream classification tasks (fine-tuning). The authors propose algorithms for unlearning documents in both the pretraining and fine-tuning phases and prove that the models output by these algorithms have similar performance to, and are nearly indistinguishable from, models that were never trained on the unlearned documents. The results also show that unlearning is "easier" (more documents can be unlearned with the same degradation in task performance) in the fine-tuning phase. The key technique in the proof (as I understand it) is to show that the estimates of word co-occurrence statistics in terms of "anchor word" co-occurrence vectors don't change much when updated by the unlearning algorithm.

**Strengths:**

- Serious theoretical effort to give provable unlearning guarantees in a simplified setting. An important step towards understanding unlearning for more complicated pretrain/finetune setups.

- Builds on the provable topic modeling results of Arora et al. (2012) to give much stronger unlearning guarantees than other work on provable unlearning.

- The main results are clearly explained and contextualized, especially the difference between unlearning for pretrained models and unlearning for fine-tuned models.

**Weaknesses:**

- The presentation is a bit hard to follow and relies heavily on many results from Arora et al. (2012).
  - $(\epsilon, \delta)$-unlearning is mentioned but not defined in L85--86. It would be helpful for those unfamiliar w/ the formalization of unlearning to at least sketch the details here so the reader can understand the type of guarantee that the paper is trying to give.

  - The need for a utility-preserving unlearning definition is not well-motivated. In Definition 4, it seems intuitive that (1) should imply that the unlearned model is not much worse than the output of $\mathcal{A}(S)$ except in pathological cases.

  - Related to the previous point, why does (2) compare to the irreducible loss (i.e., the best-achievable loss) $h^*$ and not the loss of the model output by $\mathcal{A}(S)$? "Utility-preserving" intuitively sounds like unlearning from a model $m = \mathcal{A}(S)$ gives me a new model $m'$ that is not much worse than $m$. But instead this definition compares to the loss of the optimal model $m^*$. Doesn't this essentially pre-suppose that $\mathcal{A}(S)$ is a very good learner?

  - In Lemma 1, $(\epsilon, \delta)$-differentially-private is not defined. Can the authors relate it back to Definition 3?

  - What is the intuition for $\gamma$ in Theorem 1?

  - The stated results don't indicate how the unlearning capacity depends on the minimum anchor word conditional probability $p,$ but clearly this is a very important quantity. What if my unlearning set contains all the samples with that topic's anchor word? (It's clear in the proof, but maybe it'd be better to make it clear in the Theorem statement)

  - Important quantities, such as $C^{F}$, are never defined in the text and only explained in pseudocode in Algorithm 1.
  - A numerical example / walkthrough of the proposed algorithm on a synthetic example would be very helpful for understanding. It's currently easy to get lost in the notation

- The results would be stronger if the authors connected them more to empirical results in other settings (beyond topic models). For example, they show that unlearning is easier for task-specific models than pretraining. Is this backed up by empirical evidence with more realistic models?

**Questions:**

- other questions above

- "it is unclear why one would want to use A without w or vice versa." I agree $w$ isn't useful without $A$, but isn't $A$ useful without $w$? It seems plausible that $A$ and $w$ would be released separately so $A$ can be used some other unforseen tasks.

- For related work on "Theoretical analysis of pre-training and fine-tuning", the authors might be interested in work that studies provable guarantees for contrastive pretraining and head tuning in the context of topic models: https://www.jmlr.org/papers/volume22/21-0089/21-0089.pdf

---

> ### Author Response · Authors · 2024-11-19
> **Response to Reviewer Hp5C (Part 1)**
>
> Thank you for the detailed feedback on our paper and for the pointer to related work! We address your concerns below, and please let us know if you have further questions or suggestions for our paper.
>
> > **Q1: The need for a utility-preserving unlearning definition is not well-motivated. In Definition 4, it seems intuitive that (1) should imply that the unlearned model is not much worse than the output of A(S) except in pathological cases.**
>
> **A1**: (1) is a statement that only concerns the indistinguishability between the unlearned model and the retrained model on $S \setminus S_f$. In particular, it does not say anything about the performance of the unlearned model, which depends on several other factors. For example, it may be possible that the unlearning algorithm can match the retrained model even after unlearning half of the dataset, but suffer a large cost in accuracy. Even for smaller unlearning sets, it may be the case that the unlearned model is close to the retrained model but suffers in accuracy for certain worst case small unlearning sets. Overall, (1) outlines the criteria for unlearning and (2) dictates the need for having a useful model after unlearning. The utility preserving unlearning definition aims to capture these two criteria within one comprehensive definition.
>
> > **Q2: Related to the previous point, why does (2) compare to the irreducible loss (i.e., the best-achievable loss) h∗ and not the loss of the model output by A(S)? "Utility-preserving" intuitively sounds like unlearning from a model m=A(S) gives me a new model m′ that is not much worse than m. But instead this definition compares to the loss of the optimal model m∗. Doesn't this essentially pre-suppose that A(S) is a very good learner?**
>
> **A2**: In the realizable setting, the irreducible loss is 0, as the best model is simply the ground truth topic model $\mathbf{A}^\star$. In this sense, the best utility is achieved by using $\mathbf{A}^\star$, and is what we compare to in (2), rather than $\mathcal{A}(S)$, because while $\mathcal{A}(S)$ is a learned model that can learn arbitrarily close to the $\mathbf{A}^\star$ with increasing number of samples used for training, it is still not 100% perfect in utility. By the learning algorithm guarantee in Theorem 1, $\mathcal{A}(S)$ is in fact a good learner, which is a prerequisite to the unlearning algorithm (otherwise we have no utility that we need to preserve when we unlearn).
>
> > **Q3: In Lemma 1, (ϵ,δ)-differentially-private is not defined. Can the authors relate it back to Definition 3?**
>
> **A3**: We thank the reviewer for pointing this out and we will include a note on this in the revised version. For our purposes, we can think of differential privacy interchangeably with model indistinguishability. To clarify, the standard definition of (ϵ,δ)-differentially private is essentially equivalent to having functions of adjacent datasets (i.e. datasets who differ by one data point) be (ϵ,δ)-indistinguishable [2]. In our unlearning definition (as well as other unlearning definitions in the literature), the analogue of adjacent datasets can be interpreted as the full dataset $S$ and the retain dataset $S \setminus S_f$. Specifically, the unlearning definition requires (ϵ,δ)-indistinguishability between unlearning $S_f$ from $\mathcal{A}(S)$, and unlearning the empty set from $\mathcal{A}(S \setminus S_f)$.
>
> > **Q4: What is the intuition for γ in Theorem 1?**
>
> **A4**: In Theorem 1 (as given by [1]), γ can be thought of as how “robust” the simplex of the ground truth word-word co-occurrence is, in the sense of reasoning about how much the simplex changes upon removal of any simplex vertex. Slightly more formally, γ is a lower bound on how far the removed vertex would be from the new simplex without it; therefore, one would intuitively expect an inverse dependence of γ in the sample complexity of the learning algorithm, as smaller γ would mean more samples are needed to “detect” this vertex.
>
> > **Q5: The stated results don't indicate how the unlearning capacity depends on the minimum anchor word conditional probability p, but clearly this is a very important quantity. What if my unlearning set contains all the samples with that topic's anchor word? (It's clear in the proof, but maybe it'd be better to make it clear in the Theorem statement)**
>
> **A5**: This is a valid question, and in our theorem statement, we have hidden this value in the constants of our theorem, since it is a quantity that we cannot control, but is rather an underlying property of the topic model. For the exact dependency, refer to equation 58 in the appendix, located in the proof of Lemma 18. Nevertheless, while it is important, we choose to highlight the other important quantities that govern the asymptotic deletion capacity with respect to downstream tasks, namely the number of topics. We will clarify this in the text of the main paper.

---

> ### Author Response · Authors · 2024-11-19
> **Response to Reviewer Hp5C (Part 2)**
>
> > **Q6: The results would be stronger if the authors connected them more to empirical results in other settings (beyond topic models). For example, they show that unlearning is easier for task-specific models than pretraining. Is this backed up by empirical evidence with more realistic models?**
>
> **A6**: Our observation that unlearning becomes easier for task-specific models is an interesting phenomenon that we hope will motivate future empirical work in this domain. We are currently not aware of empirical evidence on this specific phenomenon; however, there are many unlearning works that employ downstream tasks as a means of evaluating unlearning in large language models [3, 4]. More importantly, we show a theoretical example of the pretraining/fine-tuning paradigm in which we do not even update the base model parameter weights for unlearning, something that has been increasingly popular lately [5, 6]. As for the difficulty of unlearning in task-specific settings, one would imagine that task-specialized unlearning is less stringent than task-agnostic unlearning; to that end, our theory serves to formalize this through a rigorous analysis of the unlearning process, which is one of the key motivations for our work.
>
> > **Q7: I agree w isn't useful without A, but isn't A useful without w? It seems plausible that A and w would be released separately so A can be used some other unforseen tasks.**
>
> **A7**: Yes, $\mathbf{A}$ can be useful for other tasks, which is why we give an unlearning algorithm and its corresponding deletion capacity for just the base model (Theorem 2). If one is releasing $\mathbf{A}$ on its own, they can follow the protocol there. However, it is most common to fine-tune topic models for downstream tasks, and so we give additional guarantees when we want to use $\mathbf{A}$ in a downstream setting (in which we do not care about $\mathbf{A}$ individually but instead $\mathbf{A}\mathbf{w}$ as a whole). The increased deletion capacity in the fine-tuning setting may inspire model owners to rethink what types of models they release.
>
> ---
> Once again, we thank you for the feedback on the clarity and presentation - we will take these points into account for future revisions of the paper!
>
> References:
>
> [1] Sanjeev Arora, Rong Ge, Yonatan Halpern, David M. Mimno, Ankur Moitra, David Sontag, Yichen Wu, and Michael Zhu. 2013. A practical algorithm for topic modeling with provable guarantees. In *Proceedings of International Conference on Machine Learning*, pages 280–288.
>
> [2] Cynthia Dwork, Aaron Roth, et al. The algorithmic foundations of differential privacy. *Foundations
> and Trends® in Theoretical Computer Science*, 9(3–4):211–407, 2014.
>
> [3] Weijia Shi, Anirudh Ajith, Mengzhou Xia, Yangsibo Huang, Daogao Liu, Terra Blevins, Danqi
> Chen, and Luke Zettlemoyer. Detecting pretraining data from large language models. *arXiv
> preprint arXiv:2310.16789*, 2023.
>
> [4] Aengus Lynch, Phillip Guo, Aidan Ewart, Stephen Casper, and Dylan Hadfield-Menell. Eight
> methods to evaluate robust unlearning in llms. *arXiv preprint arXiv:2402.16835*, 2024
>
> [5] Martin Pawelczyk, Seth Neel, and Himabindu Lakkaraju. In-context unlearning: Language models
> as few shot unlearners. *arXiv preprint arXiv:2310.07579*, 2023.
>
> [6] Karuna Bhaila, Minh-Hao Van, and Xintao Wu. Soft prompting for unlearning in large language
> models. *arXiv preprint arXiv:2406.12038*, 2024.

---

### Official Review · Reviewer_LuZt · 2024-11-04

**Soundness:** 3
**Presentation:** 3
**Contribution:** 3
**Rating:** 6
**Confidence:** 4

**Summary:**

This paper studies machine unlearning in topic models and downstream tasks. The authors developed unlearning algorithms that enjoys provable guarantees, for both the topic model and downstream tasks. The proposed unlearning algorithms can be implemented efficiently, with runtime independent of the dataset size.

**Strengths:**

The authors developed unlearning algorithms with provable guarantees for topic models and its downstream tasks. This can be viewed as the first theoretical guarantees for unlearning in the pretraining and finetuning paradigm. Also, the proposed unlearning algorithms can be implemented efficiently, with runtime independent of the dataset size.

**Weaknesses:**

1. It seems that the proposed algorithms are heavily related to the learning algorithm developed by Arora et al. 2012. Can authors generalize their results to other learning algorithms?
2. Since this is the first unlearning algorithm in the pretraining and finetuning paradigm that enjoys provable guarantees, can authors highlight their technical contributions in deriving the results? What are the main breakthroughs compared to developing provable unlearning algorithms in the supervised setting?
3. There are several constants in Definition 4 (e.g., 0.9 and 0.01). Is that possible to make these constants more universal, e.g., to $\gamma_1$ and $\gamma_2$? If so, how would $\gamma_1$ and $\gamma_2$ affect the developed guarantees?

**Questions:**

See the Weaknesses section.

---

> ### Author Response · Authors · 2024-11-19
> **Response to Reviewer LuZt**
>
> Thank you for your valuable feedback and questions! Below, we address the aforementioned concerns, and do let us know if you have any further questions.
>
> > **Q1: It seems that the proposed algorithms are heavily related to the learning algorithm developed by Arora et al. 2012.**
>
> **A1**: This is true, but it is necessary for any unlearning algorithm to be dependent on the choice of learning algorithm. The goal is to approximate the model that would have resulted if we had omitted this datapoint during training, and so the choice of learning algorithm is fundamentally linked to the design of the unlearning algorithm. In other words, this is a standard feature of unlearning algorithms with provable guarantees.
>
> > **Q2: Since this is the first unlearning algorithm in the pretraining and finetuning paradigm that enjoys provable guarantees, can authors highlight their technical contributions in deriving the results? What are the main breakthroughs compared to developing provable unlearning algorithms in the supervised setting?**
>
> **A2**: The key intuition is that the re-trained model and the unlearned model only need to behave identically on the downstream task, which covers a narrower set of topics than pre-training. In particular, the $q$ value in Theorem 4 quantifies the extent of feature extraction required by the task. When $q$ is large, there are less features/topics that need to be extracted, which give the improved deletion capacity, and the opposite holds when $q$ is small.
>
> We also discuss here how our analysis differs from the one in the supervised setting. The supervised setting requires computing the influence of a single training data point (for some definition of influence), and surgically removing that. In our work, the influence of a single data training document is less direct due to the feature learning aspect in the pipeline, and so we need to understand how the training algorithm uses a specific document in order to remove the effect of that document. As we described in Section 4, the learning algorithm requires constructing a convex hull of the topics in the document, and the effect of one document on this hull is not as straightforward as a simple influence function.
>
> > **Q3: There are several constants in Definition 4 (e.g., 0.9 and 0.01). Is that possible to make these constants more universal, e.g., to γ1 and γ2? If so, how would γ1 and γ2 affect the developed guarantees?**
>
> **A3**: The exact numbers actually do not matter too much; as long as they are constants, they do not affect our asymptotic results. In particular, we simply choose 0.9 and 0.01 for simplicity, and regardless, the guarantees in the relevant theorems will still hold as is. This is also the same way that [1] had previously defined these quantities.
>
> References:
>
> [1] Ayush Sekhari, Jayadev Acharya, Gautam Kamath, and Ananda Theertha Suresh. Remember what you want to forget: Algorithms for machine unlearning. *Advances in Neural Information Processing Systems*, 34:18075–18086, 2021.

---

> > ### Comment · Reviewer_LuZt · 2024-11-24
> >
> > Thank you for your response. I'd like to keep my rating.

---

### Meta-Review · Area_Chair_yMxZ · 2024-12-21

**Metareview:**

The paper considers the problem of unlearning topic models which are learned using the topic modeling algorithm of Arora et a. 2012. The reviewers found the paper technically solid. The reviewers had concerns that the problem setting was a bit too specialized (topic models), or that the paper considers a specific algorithm for the task (one in Arora et a. 2012, as the authors say some assumption about the learning algorithm is necessary but the choice is potentially a bit arbitrary and some more generality could be good, why not even consider something simpler like SVD?). However, since the work makes good technical progress on a relevant problem (unlearning in the context of language/topic models), I recommend acceptance. I suggest that the authors try to rectify some of the above issues in the revision of the paper.

**Additional Comments On Reviewer Discussion:**

There was a decent amount of discussion on this, and the authors tried to address the concerns regarding the topic modeling setting. The reviewers were mostly not that convinced about the connection to language models, but agreed that the work is solid and relevant, even if in isolation.

---

### Decision · Program_Chairs · 2025-01-22

Accept (Poster)